# Weisfeiler and Leman Go Relational

**Pablo Barcelo**[*]

Institute for Mathematical and Computational Engineering, PUC Chile & IMFD Chile & CENIA Chile
pbarcelo@uc.cl

**Mikhail Galkin**
Mila Quebec AI Institute & McGill University
mikhail.galkin@mila.quebec

**Christopher Morris**
RWTH Aachen University
morris@cs.rwth-aachen.de

**Miguel Romero**
Universidad Adolfo Ibáñez & CENIA Chile
miguel.romero.o@uai.cl

## Abstract

Knowledge graphs, modeling multi-relational data, improve numerous applications such as question answering or graph logical reasoning. Many graph neural networks for such data emerged recently, often outperforming shallow architectures. However, the design of such multi-relational graph neural networks is ad-hoc, driven mainly by intuition and empirical insights. Up to now, their expressivity, their relation to each other, and their (practical) learning performance is poorly understood. Here, we initiate the study of deriving a more principled understanding of multi-relational graph neural networks. Namely, we investigate the limitations in the expressive power of the well-known Relational GCN and Compositional GCN architectures and shed some light on their practical learning performance. By aligning both architectures with a suitable version of the Weisfeiler-Leman test, we establish under which conditions both models have the same expressive power in distinguishing non-isomorphic (multi-relational) graphs or vertices with different structural roles. Further, by leveraging recent progress in designing expressive graph neural networks, we introduce the $k$-RN architecture that provably overcomes the expressiveness limitations of the above two architectures. Empirically, we confirm our theoretical findings in a vertex classification setting over small and large multi-relational graphs.

## 1 Introduction

Recently, GNNs (Gilmer et al., 2017; Scarselli et al., 2009) emerged as the most prominent graph representation learning architecture. Notable instances of this architecture include, e.g., Duvenaud et al. (2015); Hamilton et al. (2017), and Veličković et al. (2018), which can be subsumed under the message-passing framework introduced in Gilmer et al. (2017). In parallel, approaches based on spectral information were introduced in, e.g., Defferrard et al. (2016); Bruna et al. (2014); Kipf and Welling (2017), and Monti et al. (2017)—all of which descend from early work in Scarselli et al. (2009); Baskin et al. (1997); Kireev (1995); Micheli and Sestito (2005); Merkwirth and Lengauer (2005); Micheli (2009) and Sperduti and Starita (1997).

By now, we have a deep understanding of the expressive power of GNNs (Morris et al., 2021). To start with, connections between GNNs and Weisfeiler–Leman type algorithms have been shown. Specifically, Morris et al. (2019) and Xu et al. (2019) showed that the 1-WL limits the expressive power of any possible GNN architecture in terms of distinguishing non-isomorphic graphs. In turn,

---

[*]Alphabetical author order.

Barcelo et al., Weisfeiler and Leman Go Relational. *Proceedings of the First Learning on Graphs Conference (LoG 2022)*, PMLR 198, Virtual Event, December 9–12, 2022.

these results have been generalized to the $k$-WL, see, e.g., Morris et al. (2019); Azizian and Lelarge (2020); Geerts et al. (2020); Geerts (2020); Maron et al. (2019); Morris et al. (2020, 2022a), and connected to permutation-equivariant function approximation over graphs, see, e.g., Chen et al. (2019); Geerts and Reutter (2022); Maehara and NT (2019). Barceló et al. (2020) further established an equivalence between the expressiveness of GNNs with readout functions and $C^2$, the 2-variable fragment of first-order logic with counting quantifiers.

Most previous works focus on graphs that admit labels on vertices but not edges. However, *knowledge* or *multi-relational graphs*, that admit labels on both vertices and edges play a crucial role in numerous applications, such as complex question answering in NLP (Fu et al., 2020) or visual question answering (Huang et al., 2022) in the intersection of NLP and vision. To extract the rich information encoded in the graph's multi-relational structure and its annotations, the knowledge graph community has proposed a large set of *relational* GNN architectures, e.g., Schlichtkrull et al. (2018); Vashishth et al. (2020); Ye et al. (2022), tailored towards knowledge or multi-relational graphs, targeting tasks such as vertex and link prediction (Schlichtkrull et al., 2018; Ye et al., 2022; Zhu et al., 2021). Notably, Schlichtkrull et al. (2018) proposed the first architecture, namely, R-GCN, being able to handle multi-relational data. Further, Vashishth et al. (2020) proposed an alternative GNN architecture, CompGCN, using less number of parameters and reported improved empirical performance. In the knowledge graph reasoning area, R-GCN and CompGCN, being strong baselines, spun off numerous improved GNNs for vertex classification and transductive link prediction tasks (Galkin et al., 2020; Yu et al., 2020; Zhang et al., 2022). They also inspired architectures for more complex reasoning tasks such as inductive link prediction (Zhu et al., 2021; Teru et al., 2020; Ali et al., 2021a; Zhang and Yao, 2022) and query answering (Daza and Cochez, 2020; Alivanistos et al., 2022; Zhu et al., 2022).

Although these approaches show meaningful empirical performance, their limitations in extracting relevant structural information, their learning performance, and their relation to each other are not understood well. For example, there is no understanding of these approaches' inherent limitations in distinguishing between knowledge graphs with different structural features, explicitly considering the unique properties of multi-relational graphs. Hence, a thorough theoretical investigation of multi-relational GNNs' expressive power and learning performance is yet to be established to become meaningful, vital components in today's knowledge graph reasoning pipeline.

**Present Work.** Here, we initiate the study on deriving a principled understanding of the capabilities of GNNs for knowledge or multi-relational graphs. More concretely:

- We investigate the expressive power of two well-known GNNs for multi-relation data, *Relational GCNs* (R-GCN) (Schlichtkrull et al., 2018) and *Compositional GCNs* (CompGCN) (Vashishth et al., 2020). We quantify their limitations by relating them to a suitable version of the established Weisfeiler-Leman graph isomorphism test. In particular, we show under which conditions the above two architectures possess the same expressive power in distinguishing non-isomorphic, multi-relational graphs or vertices with different structural features.

- To overcome both architectures' expressiveness limitations, we introduce the $k$-RN architecture, which provably overcomes their limitations and show that increasing $k$ always leads to strictly more expressive architectures.

- Empirically, we confirm our theoretical findings on established small- and large-scale multi-relational vertex classification benchmarks.

See Subsection A.1 in the appendix for an expanded discussion of related work.

## 2 Preliminaries

As usual, let $[n] = \{1, \ldots, n\} \subset \mathbb{N}$ for $n \geq 1$, and let $\{\!\!\{ \ldots \}\!\!\}$ denote a multiset.

A *(undirected) graph* $G$ is a pair $(V(G), E(G))$ with a *finite* set of *vertices* $V(G)$ and a set of *edges* $E(G) \subseteq \{\{u, v\} \subseteq V \mid u \neq v\}$. For notational convenience, we usually denote an edge $\{u, v\}$ in $E(G)$ by $(u, v)$ or $(v, u)$. We assume the usual definition of *adjacency matrix* $\boldsymbol{A}$ of $G$. A *colored* or *labeled graph* $G$ is a triple $(V(G), E(G), \ell)$ with a *coloring* or *label* function $\ell \colon V(G) \to \mathbb{N}$. Then $\ell(w)$ is a *color* or *label* of $w$, for $w$ in $V(G)$. The *neighborhood* of $v$ in $V(G)$ is denoted by $N(v) = \{u \in V(G) \mid (v, u) \in E(G)\}$.

An *(undirected) multi-relational graph* $G$ is a tuple $(V(G), R_1(G), \ldots, R_r(G))$ with a *finite* set of *vertices* $V(G)$ and *relations* $R_i \subseteq \{\{u, v\} \subseteq V(G) \mid u \neq v\}$ for $i$ in $[r]$. The *neighborhood* of $v$ in $V(G)$ with respect to the relation $R_i$ is denoted by $N_i(v) = \{u \in V(G) \mid (v, u) \in R_i\}$. We define *colored* (or *labeled*) multi-relational graphs in the expected way.

Two graphs $G$ and $H$ are *isomorphic* ($G \simeq H$) if there exists a bijection $\varphi \colon V(G) \to V(H)$ preserving the adjacency relation, i.e., $(u, v)$ in $E(G)$ if and only if $(\varphi(u), \varphi(v))$ in $E(H)$. We then call $\varphi$ an *isomorphism* from $G$ to $H$. If the graphs have vertex labels, the isomorphism is additionally required to match these labels. In the case of multi-relational graphs $G$ and $H$, the bijection $\varphi \colon V(G) \to V(H)$ needs to preserve all relations, i.e., $(u, v)$ is in $R_i(G)$ if and only if $(\varphi(u), \varphi(v))$ is in $R_i(H)$ for each $i$ in $[r]$. For labeled multi-relational graphs, the bijection needs to preserve the labels.

We define the atomic type $\mathsf{atp} \colon V(G)^k \to \mathbb{N}$ such that $\mathsf{atp}(\mathbf{v}) = \mathsf{atp}(\mathbf{w})$ for $\mathbf{v}$ and $\mathbf{w}$ in $V(G)^k$ if and only if the mapping $\varphi \colon V(G) \to V(G)$ where $v_i \mapsto w_i$ induces a *partial isomorphism*, i.e., $v_i = v_j \iff w_i = w_j$ and $(v_i, v_j)$ in $E(G) \iff (\varphi(v_i), \varphi(v_j))$ in $E(G)$.

**The Weisfeiler-Leman Algorithm.** The 1-*dimensional Weisfeiler-Leman algorithm* (1-WL), or *color refinement*, is a simple heuristic for the graph isomorphism problem, originally proposed by Weisfeiler and Leman (1968).[2][3] Intuitively, the algorithm determines if two graphs are non-isomorphic by iteratively coloring or labeling vertices. Given an initial coloring or labeling of the vertices of both graphs, e.g., their degree or application-specific information, in each iteration, two vertices with the same label get different labels if the number of identically labeled neighbors is not equal. If, after some iteration, the number of vertices annotated with a specific label is different in both graphs, the algorithm terminates and a stable coloring, inducing a vertex partition, is obtained. We can then conclude that the two graphs are not isomorphic. It is easy to see that the algorithm cannot distinguish all non-isomorphic graphs (Cai et al., 1992a). Nonetheless, it is a powerful heuristic that can successfully test isomorphism for a broad class of graphs (Arvind et al., 2015; Babai and Kucera, 1979; Kiefer et al., 2015).

Formally, let $G = (V(G), E(G), \ell)$ be a labeled graph. In each iteration, $t > 0$, the 1-WL computes a vertex coloring $C^{(t)} \colon V(G) \to \mathbb{N}$, which depends on the coloring of the neighbors. That is, in iteration $t > 0$, we set

$$C^{(t)}(v) := \mathsf{RELABEL}\Big(\big(C^{(t-1)}(v), \{\!\!\{C^{(t-1)}(u) \mid u \in N(v)\}\!\!\}\big)\Big),$$

where RELABEL injectively maps the above pair to a unique natural number, which has not been used in previous iterations. In iteration 0, the coloring $C^{(0)} := \ell$. To test if two graphs $G$ and $H$ are non-isomorphic, we run the above algorithm in "parallel" on both graphs. If the two graphs have a different number of vertices colored $c$ in $\mathbb{N}$ at some iteration, the 1-WL *distinguishes* the graphs as non-isomorphic. Moreover, if the number of colors between two iterations, $t$ and $(t + 1)$, does not change, i.e., the cardinalities of the images of $C^{(t)}$ and $C^{(t+1)}$ are equal, or, equivalently,

$$C^{(t)}(v) = C^{(t)}(w) \iff C^{(t+1)}(v) = C^{(t+1)}(w),$$

for all vertices $v$ and $w$ in $V(G)$, the algorithm terminates. For such $t$, we define the *stable coloring* $C^{\infty}(v) = C^{(t)}(v)$ for $v$ in $V(G)$. The stable coloring is reached after at most $\max\{|V(G)|, |V(H)|\}$ iterations (Grohe, 2017).

Due to the shortcomings of the 1-WL in distinguishing non-isomorphic graphs, several researchers, e.g., (Babai, 1979; Cai et al., 1992b), devised a more powerful generalization of the former, today known as the $k$-dimensional Weisfeiler-Leman algorithm ($k$-WL), see Subsection A.2 for details.

**Graph Neural Networks.** Intuitively, GNNs learn a vectorial representation, i.e., a $d$-dimensional vector, representing each vertex in a graph by aggregating information from neighboring vertices. Formally, let $G = (V(G), E(G), \ell)$ be a labeled graph with initial vertex features $(\boldsymbol{h}_v^{(0)})_{v \in V(G)}$ in

---

[2]Strictly speaking, 1-WL and color refinement are two different algorithms. That is, 1-WL considers neighbors and non-neighbors to update the coloring, resulting in a slightly higher expressive power when distinguishing vertices in a given graph, see Grohe (2021) for details. For brevity, we consider both algorithms to be equivalent.

[3]We use the spelling "Leman" here as A. Leman, co-inventor of the algorithm, preferred it over the transcription "Lehman"; see `https://www.iti.zcu.cz/wl2018/pdf/leman.pdf`.

$\mathbb{R}^d$ that are *consistent* with $\ell$, that is, $\boldsymbol{h}_u^{(0)} = \boldsymbol{h}_v^{(0)}$ if and only if $\ell(u) = \ell(v)$, e.g., a one-hot encoding of the labelling $\ell$. Alternatively, $(\boldsymbol{h}_v^{(0)})_{v \in V(G)}$ can be arbitrary vertex features annotating the vertices of $G$.

A GNN architecture consists of a stack of neural network layers, i.e., a composition of permutation-invariant or -equivariant parameterized functions. Similarly to 1-WL, each layer aggregates local neighborhood information, i.e., the neighbors' features, around each vertex and then passes this aggregated information on to the next layer.

GNNs are often realized as follows (Morris et al., 2019). In each layer, $t > 0$, we compute vertex features

$$\boldsymbol{h}_v^{(t)} := \sigma\Big(\boldsymbol{h}_v^{(t-1)}\boldsymbol{W}_0^{(t)} + \sum_{w \in N(v)} \boldsymbol{h}_w^{(t-1)}\boldsymbol{W}_1^{(t)}\Big) \in \mathbb{R}^e, \tag{1}$$

for $v$ in $V(G)$, where $\boldsymbol{W}_0^{(t)}$ and $\boldsymbol{W}_1^{(t)}$ are parameter matrices from $\mathbb{R}^{d \times e}$ and $\sigma$ denotes an entry-wise non-linear function, e.g., a sigmoid or a ReLU function.[4] Following Gilmer et al. (2017) and Scarselli et al. (2009), in each layer, $t > 0$, we can generalize the above by computing a vertex feature

$$\boldsymbol{h}_v^{(t)} := \mathsf{UPD}^{(t)}\Big(\boldsymbol{h}_v^{(t-1)}, \mathsf{AGG}^{(t)}\big(\{\!\!\{\boldsymbol{h}_w^{(t-1)} \mid w \in N(v)\}\!\!\}\big)\Big),$$

where $\mathsf{UPD}^{(t)}$ and $\mathsf{AGG}^{(t)}$ may be differentiable parameterized functions, e.g., neural networks.[5] In the case of graph-level tasks, e.g., graph classification, one uses

$$\boldsymbol{h}_G := \mathsf{READOUT}\big(\{\!\!\{\boldsymbol{h}_v^{(T)} \mid v \in V(G)\}\!\!\}\big),$$

to compute a single vectorial representation based on learned vertex features after iteration $T$. Again, READOUT may be a differentiable parameterized function. To adapt the parameters of the above three functions, they are optimized end-to-end, usually through a variant of stochastic gradient descent, e.g., (Kingma and Ba, 2015), together with the parameters of a neural network used for classification or regression.

**Graph Neural Networks for Multi-relational Graphs.** In the following, we describe GNN layers for multi-relational graphs, namely R-GCN (Schlichtkrull et al., 2018) and CompGCN (Vashishth et al., 2020). Initial features are computed in the same way as in the previous subsection.

**R-GCN.** Let $G$ be a labeled multi-relational graph. In essence, R-GCN generalizes Equation 1 by using an additional sum iterating over the different relations. That is, we compute a vertex feature

$$\boldsymbol{h}_{v,\mathsf{R\text{-}GCN}}^{(t)} := \sigma\Big(\boldsymbol{h}_{v,\mathsf{R\text{-}GCN}}^{(t-1)}\boldsymbol{W}_0^{(t)} + \sum_{i \in [r]} \sum_{w \in N_i(v)} \boldsymbol{h}_{w,\mathsf{R\text{-}GCN}}^{(t-1)}\boldsymbol{W}_i^{(t)}\Big) \in \mathbb{R}^e, \tag{2}$$

for $v$ in $V(G)$, where $\boldsymbol{W}_0^{(t)}$ and $\boldsymbol{W}_i^{(t)}$ for $i$ in $[r]$ are parameter matrices from $\mathbb{R}^{d \times e}$, and $\sigma$ denotes a entry-wise non-linear function. We note here that the original R-GCN layer defined in Schlichtkrull et al. (2018) uses a mean operation instead of a sum in the most inner sum of Equation 2. We investigate the empirical advantages of these two variations in Section 5.

**CompGCN.** Let $G$ be a labeled multi-relational graph. A CompGCN layer generalizes Equation 1 by encoding relational information as edge features. That is, we compute a vertex feature

$$\boldsymbol{h}_{v,\mathsf{CompGCN}}^{(t)} := \sigma\Big(\boldsymbol{h}_{v,\mathsf{CompGCN}}^{(t-1)}\boldsymbol{W}_0^{(t)} + \sum_{i \in [r]} \sum_{w \in N_i(v)} \phi\big(\boldsymbol{h}_{w,\mathsf{CompGCN}}^{(t-1)}, \boldsymbol{z}_i^{(t)}\big)\boldsymbol{W}_1^{(t)}\Big) \in \mathbb{R}^e, \tag{3}$$

for $v$ in $V(G)$, where $\boldsymbol{W}_0^{(t)}$ and $\boldsymbol{W}_1^{(t)}$ are parameter matrices from $\mathbb{R}^{d \times e}$ and $\mathbb{R}^{c \times e}$, respectively, and $\boldsymbol{z}_i^{(t)}$ in $\mathbb{R}^b$ is the learned edge feature for the $i$th relation at layer $t$. Further, the function $\phi \colon \mathbb{R}^d \times \mathbb{R}^b \to \mathbb{R}^c$ is a *composition map*, mapping two vectors onto a single vector in a non-parametric way, e.g., summation, point-wise multiplication, or concatenation. We note here that the original CompGCN layer defined in Vashishth et al. (2020) uses an additional sum to differentiate between in-going and out-going edges and self loops, see Section E for details.

---

[4]For clarity of presentation, we omit biases.

[5]Strictly speaking, Gilmer et al. (2017) consider a slightly more general setting in which vertex features are computed by $\boldsymbol{h}_v^{(t+1)} := \mathsf{UPD}^{(t+1)}\Big(\boldsymbol{h}_v^{(t)}, \mathsf{AGG}^{(t+1)}\big(\{\!\!\{(\boldsymbol{h}_v^{(t)}, \boldsymbol{h}_w^{(t)}, \ell(v,w)) \mid w \in N(v)\}\!\!\}\big)\Big)$.

## 3 Relational Weisfeiler–Leman Algorithm

In the following, to study the limitations in expressivity of the above two GNN layers, R-GCN and CompGCN, we define the *multi-relational* 1-*WL* (1-RWL). Let $G = (V(G), R_1(G), \ldots, R_r(G), \ell)$ be a labeled, multi-relational graph. Then the 1-RWL computes a vertex coloring $C_{\mathsf{R}}^{(t)} \colon V(G) \to \mathbb{N}$ for $t > 0$ by interpreting the different relations as edge types, i.e.,

$$C_{\mathsf{R}}^{(t)}(v) \coloneqq \mathsf{RELABEL}\Big(\big(C_{\mathsf{R}}^{(t-1)}(v), \{\!\{(C_{\mathsf{R}}^{(t-1)}(u), i) \mid i \in [r], u \in N_i(v)\}\!\}\big)\Big), \tag{4}$$

for $v$ in $V(G)$. In iteration 0, the coloring $C_{\mathsf{R}}^{(0)} \coloneqq \ell$. In particular, two vertices $v$ and $w$ of the same color in iteration $(t-1)$ get different colors in iteration $t$ if there is a relation $R_i$ such that the number of neighbors in $N_i(v)$ and $N_i(w)$ colored with a certain color is different. We define the stable coloring $C_{\mathsf{R}}^{\infty}$ in the expected way, analogously to the 1-WL.

**Relationship Between** 1-**WL and** 1-**RWL.** Since 1-WL does not consider edge labels it is clear that 1-RWL is strictly stronger than the 1-WL. For example, take a pair of isomorphic graphs and label the edges differently in each graphs, making the graph non-isomorphic. Clearly, 1-RWL will distinguish them while 1-WL will not.

**Relationship Between** 1-**RWL, R-GCN, and CompGCN.** Morris et al. (2019); Xu et al. (2019) established the exact relationship between the expressive power of 1-WL and GNNs. In particular, 1-WL upper bounds the capacity of any GNN architecture for distinguishing vertices in graphs. In turn, over every graph $G$ there is a GNN architecture with the same expressive power as 1-WL for distinguishing vertices in $G$. In this section, we show that the same relationship can be established between multi-relational 1-WL, on the one hand, and the R-GCN and CompGCN architectures, on the other.

Let $G = (V(G), R_1(G), \ldots, R_r(G), \ell)$ be a labeled, multi-relational graph, and let

$$\mathbf{W}_{\mathsf{R\text{-}GCN}}^{(t)} = \big(\boldsymbol{W}_0^{(t')}, \boldsymbol{W}_i^{(t')}\big)_{t' \leq t, i \in [r]}$$

denote the sequence of R-GCN parameters given by Equation 2 up to iteration $t$. Analogously, we denote by

$$\mathbf{W}_{\mathsf{CompGCN}}^{(t)} = \big(\boldsymbol{W}_0^{(t')}, \boldsymbol{W}_1^{(t')}, \boldsymbol{z}_i^{(t')}\big)_{t' \leq t, i \in [r]}$$

the sequence of CompGCN parameters given by Equation 3 up to iteration $t$. We first show that the multi-relational 1-WL upper bounds the expressivity of both the R-GCN and CompGCN layers in terms of their capacity to distinguish vertices in labeled multi-relational graphs.

**Theorem 1.** *Let $G = (V(G), R_1(G), \ldots, R_r(G), \ell)$ be a labeled, multi-relational graph. Then for all $t \geq 0$ the following holds:*

- *For all choices of initial vertex features consistent with $\ell$, sequences $\mathbf{W}_{\mathsf{R\text{-}GCN}}^{(t)}$ of R-GCN parameters, and vertices $v$ and $w$ in $V(G)$,*

$$C_{\mathsf{R}}^{(t)}(v) = C_{\mathsf{R}}^{(t)}(w) \implies \boldsymbol{h}_{v,\mathsf{R\text{-}GCN}}^{(t)} = \boldsymbol{h}_{w,\mathsf{R\text{-}GCN}}^{(t)}.$$

- *For all choices of initial vertex features consistent with $\ell$, sequences $\mathbf{W}_{\mathsf{CompGCN}}^{(t)}$ of CompGCN parameters, composition functions $\phi$, and vertices $v$ and $w$ in $V(G)$,*

$$C_{\mathsf{R}}^{(t)}(v) = C_{\mathsf{R}}^{(t)}(w) \implies \boldsymbol{h}_{v,\mathsf{CompGCN}}^{(t)} = \boldsymbol{h}_{w,\mathsf{CompGCN}}^{(t)}.$$

Noticeably, the converse also holds. That is, there is a sequence of parameter matrices $\mathbf{W}_{\mathsf{R\text{-}GCN}}^{(t)}$ such that R-GCN has the same expressive power in terms of distinguishing vertices in graphs as the coloring $C_{\mathsf{R}}^{(t)}$. This equivalence holds provided the initial labels are encoded by linearly independent vertex features, e.g., using one-hot encodings. The result also holds for CompGCN as long as the composition map $\phi$ can express vector scaling, e.g., $\phi$ is point-wise multiplication or circular correlation, two of the composition functions studied and implemented in the paper that introduced the CompGCN architecture (Vashishth et al., 2020).

**Theorem 2.** *Let $G = (V(G), R_1(G), \ldots, R_r(G), \ell)$ be a labeled, multi-relational graph. Then for all $t \geq 0$ the following holds:*

- *There are initial vertex features and a sequence $\mathbf{W}^{(t)}_{\text{R-GCN}}$ of parameters such that for all $v$ and $w$ in $V(G)$,*

$$C_R^{(t)}(v) = C_R^{(t)}(w) \iff \boldsymbol{h}^{(t)}_{v,\text{R-GCN}} = \boldsymbol{h}^{(t)}_{w,\text{R-GCN}}.$$

- *There are initial vertex features, a sequence $\mathbf{W}^{(t)}_{\text{CompGCN}}$ of parameters and a composition function $\phi$ such that for all $v$ and $w$ in $V(G)$,*

$$C_R^{(t)}(v) = C_R^{(t)}(w) \iff \boldsymbol{h}^{(t)}_{v,\text{CompGCN}} = \boldsymbol{h}^{(t)}_{w,\text{CompGCN}}.$$

**On the Choice of the Composition Function for CompGCN Architectures.** As Theorem 2 shows the expressive power of the 1-RWL is matched by that of the CompGCN architectures if we allow the latter to implement vector scaling in composition functions. However, not all composition maps that have been considered in relationship with CompGCN architectures admit such a possibility. Think, for instance, of natural composition maps such as point-wise summation or vector concatenation. Interestingly, we can show that CompGCN architectures equipped with these composition maps are provably weaker in terms of expressive power than the ones studied in the proof of Theorem 2, as they correlate with a weaker variant of 1-WL that we define next.

Let $G = (V(G), R_1(G), \ldots, R_r(G), \ell)$ be a labeled, multi-relational graph. The *weak multi-relational 1-WL* computes a vertex coloring $C_{\text{WR}}^{(t)} \colon V(G) \to \mathbb{N}$ for $t > 0$ as follows:

$$C_{\text{WR}}^{(t)}(v) := \text{RELABEL}\Big( \big( C_{\text{WR}}^{(t-1)}(v), \{\!\!\{ C_{\text{WR}}^{(t-1)}(u) \mid i \in [r], u \in N_i(v) \}\!\!\}, |N_1(v)|, \ldots, |N_r(v)| \big) \Big),$$

for $v$ in $V(G)$. In iteration 0, the coloring $C_{\text{WR}}^{(0)} := \ell$. During aggregation, the weak variant does not take information about the relations into account. The only information relative to the different relations is the number of neighbors associated with each of them. We define the stable coloring $C_{\text{WR}}^{\infty}$ analogously to the 1-WL. As it turns out, this variant is less powerful than the original one.

**Proposition 3.** *There exist a labeled, multi-relational graph $G = (V(G), R_1(G), R_2(G), \ell)$ and two vertices $v$ and $w$ in $V(G)$, such that $C_R^{(1)}(v) \neq C_R^{(1)}(w)$ but $C_{\text{WR}}^{\infty}(v) = C_{\text{WR}}^{\infty}(w)$.*

As shown next, the expressive power of CompGCN architectures that use point-wise summation or vector concatenation is captured by this weaker form of 1-RWL.

**Theorem 4.** *Let $G = (V(G), R_1(G), \ldots, R_r(G), \ell)$ be a labeled, multi-relational graph. Then for all $t \geq 0$ the following holds:*

- *For all choices of initial vertex features consistent with $\ell$, sequences $\mathbf{W}^{(t)}_{\text{CompGCN}}$ of CompGCN parameters, and vertices $v$ and $w$ in $V(G)$,*

$$C_{\text{WR}}^{(t)}(v) = C_{\text{WR}}^{(t)}(w) \implies \boldsymbol{h}^{(t)}_{v,\text{CompGCN}} = \boldsymbol{h}^{(t)}_{w,\text{CompGCN}},$$

  *for either point-wise summation or concatenation as the composition map.*

- *There exist initial vertex features and a sequence $\mathbf{W}^{(t)}_{\text{CompGCN}}$ of CompGCN parameters, such that for all vertices $v$ and $w$ in $V(G)$,*

$$C_{\text{WR}}^{(t)}(v) = C_{\text{WR}}^{(t)}(w) \iff \boldsymbol{h}^{(t)}_{v,\text{CompGCN}} = \boldsymbol{h}^{(t)}_{w,\text{CompGCN}},$$

  *for either point-wise summation or concatenation as the composition map.*

Together with Proposition 3 and Theorem 2, this result states that CompGCN architectures based on vector summation or concatenation are provably weaker in terms of their capacity to distinguish vertices in graphs than the ones that use vector scaling.

We have shown that R-GCN and CompGCN with point-wise multiplication have the same expressive power in terms of distinguishing non-isomorphic multi-relational graphs or distinguishing vertices in a multi-relational graph. As it turns out, these two architectures actually define the *same* functions. A similar result holds between CompGCN with vector summation/subtraction and concatenation. See Appendix B.2 for details.

## 4 Limitations and More Expressive Architectures

Theorem 1 shows that both R-GCN as well as CompGCN have severe limitations in distinguishing structurally different multi-relational graphs. Indeed, the following results shows that there exists pairs of non-isomorphic, multi-relational graphs that neither R-GCN nor CompGCN can distinguish.

**Proposition 5.** *For all $r \geq 1$, there exists a pair of non-isomorphic graphs $G = (V(G), R_1(G), \ldots, R_r(G), \ell)$ and $H = (V(H), R_1(H), \ldots, R_r(H), \ell)$ that cannot be distinguished by R-GCN or CompGCN.*

We note here that the two graphs $G$ and $H$ from the above theorem can also be used to show that neither R-GCN nor CompGCN will be able to compute different features for vertices in $G$ and $H$, making them indistinguishable. Hence, to overcome the limitations of the CompGCN and R-GCN, we introduce *local $k$-order relational networks* ($k$-RNs), leveraging recent progress in overcoming GNNs' inherent limitations in expressive power (Morris et al., 2021, 2019, 2020, 2022a). To do so, we first extend the local $k$-dimensional Weisfeiler–Leman algorithm (Morris et al., 2020), see Subsection A.2, to multi-relational graphs.

**Multi-relational Local $k$-WL.** Given a multi-relational graph $G = (V(G), R_1(G), \ldots, R_r(G), \ell)$, we define the *multi-relational atomic type* $\mathsf{atp}_r \colon V(G)^k \to \mathbb{N}$ such that $\mathsf{atp}_r(\mathbf{v}) = \mathsf{atp}_r(\mathbf{w})$ for $\mathbf{v}$ and $\mathbf{w}$ in $V(G)^k$ if and only if the mapping $\varphi \colon V(G) \to V(G)$ where $v_g \mapsto w_g$ induces a partial isomorphism, preserving the relations, i.e., we have $v_p = v_q \iff w_p = w_q$ and $(v_p, v_q) \in R_i(G) \iff (\varphi(v_p), \varphi(v_q)) \in R_i(G)$ for $i$ in $[r]$. The *multi-relational local $k$-WL* ($k$-RLWL) computes the coloring $C_{k,r}^{(t)} \colon V(G)^k \to \mathbb{N}$ for $t \geq 0$, where $C_{k,r}^{(0)} := \mathsf{atp}_r(\mathbf{v})$, and refines a coloring $C_{k,r}^{(t)}$ (obtained after $t$ iterations of the $k$-RLWL) via the *aggregation function*

$$
\begin{aligned}
M_r^{(t)}(\mathbf{v}) := \big( &\{\!\{(C_{k,r}^{(t)}(\theta_1(\mathbf{v}, w)), i) \mid w \in N_i(v_1) \text{ and } i \in [r]\}\!\}, \ldots, \\
&\{\!\{(C_{k,r}^{(t)}(\theta_k(\mathbf{v}, w)), i) \mid w \in N_i(v_k) \text{ and } i \in [r]\}\!\}\big),
\end{aligned}
\tag{5}
$$

where $\theta_j(\mathbf{v}, w) := (v_1, \ldots, v_{j-1}, w, v_{j+1}, \ldots, v_k)$. That is, $\theta_j(\mathbf{v}, w)$ replaces the $j$-th component of the tuple $\mathbf{v}$ with the vertex $w$. Like the local $k$-WL (Morris et al., 2020), the algorithm considers only the local $j$-neighbors, i.e., $v_i$ and $w$ must be adjacent, for each relation in each iteration and additionally differentiates between different relations. The coloring functions for the iterations of the multi-relational $k$-RLWL are then defined by

$$
C_{k,r}^{(t+1)}(\mathbf{v}) := (C_{k,r}^{(t)}(\mathbf{v}), M_r^{(t)}(\mathbf{v})).
$$

In the following, we derive a neural architecture, the $k$-RN, that has the same expressive power as the $k$-RLWL in terms of distinguishing non-isomorphic multi-relational graphs.

**The $k$-RN Architecture.** Given a labeled, multi-relational graph $G$, for each $k$-tuple $\mathbf{v}$ in $V(G)^k$, a $k$-RN architecture computes an initial feature $\boldsymbol{h}_v^{(0)}$ *consistent* with its multi-relational atomic type, e.g., a one-hot encoding of $\mathsf{atp}_r(\mathbf{v})$. In each layer, $t > 0$, a $k$-RN computes a $k$-tuple feature

$$
\begin{aligned}
\boldsymbol{h}_{\mathbf{v},k}^{(t)} := \mathsf{UPD}^{(t)}\Big( &\boldsymbol{h}_{\mathbf{v},k}^{(t-1)}, \mathsf{AGG}^{(t)}\big(\{\!\{\phi(\boldsymbol{h}_{\theta_1(\mathbf{v},w),k}^{(t-1)}, \boldsymbol{z}_i^{(t)}) \mid w \in N_i(v_1) \text{ and } i \in [r]\}\!\}, \ldots, \\
&\{\!\{\phi(\boldsymbol{h}_{\theta_k(\mathbf{v},w),k}^{(t-1)}, \boldsymbol{z}_i^{(t)}) \mid w \in N_i(v_k) \text{ and } i \in [r]\}\!\}\big)\Big) \in \mathbb{R}^e,
\end{aligned}
\tag{6}
$$

where the functions $\mathsf{UPD}^{(t)}$ and $\mathsf{AGG}^{(t)}$ for $t > 0$ may be a differentiable parameterized functions, e.g., neural networks. Similarly to Equation 3, $\boldsymbol{z}_i^{(t)}$ in $\mathbb{R}^c$ is the learned edge feature for the $i$th relation at layer $t$ and $\phi \colon \mathbb{R}^d \times \mathbb{R}^b \to \mathbb{R}^c$ is a composition map. In the case of graph-level tasks, e.g., graph classification, one uses

$$
\boldsymbol{h}_G := \mathsf{READOUT}\big(\{\!\{\boldsymbol{h}_{\mathbf{v}}^{(T)} \mid \mathbf{v} \in V(G)^k\}\!\}\big) \in \mathbb{R}^e,
\tag{7}
$$

to compute a single vectorial representation based on learned $k$-tuple features after iteration $T$. The following results shows that the $k$-RLWL upperbounds the expressivity of any $k$-RN in terms of distinguishing non-isomorphic graphs.

**Proposition 6.** *Let $G = (V(G), R_1(G), \ldots, R_r(G), \ell)$ be a labeled, multi-relational graph. Then for all $t \geq 0$, $r > 0$, $k \geq 1$, and all choices of $\mathsf{UPD}^{(t)}$, $\mathsf{AGG}^{(t)}$, and all $\mathbf{v}$ and $\mathbf{w}$ in $V(G)^k$,*

$$C_{k,r}^{(t)}(\mathbf{v}) = C_{k,r}^{(t)}(\mathbf{w}) \implies \boldsymbol{h}_{\mathbf{v},k}^{(t)} = \boldsymbol{h}_{\mathbf{w},k}^{(t)}.$$

Moreover, we can also show the converse, resulting in the following theorem.

**Proposition 7.** *Let $G = (V(G), R_1(G), \ldots, R_r(G), \ell)$ be a labeled, multi-relational graph. Then for all $t \geq 0$ and $k \geq 1$, there exists $\mathsf{UPD}^{(t)}$, $\mathsf{AGG}^{(t)}$, such that for all $\mathbf{v}$ and $\mathbf{w}$ in $V(G)^k$,*

$$C_{k,r}^{(t)}(\mathbf{v}) = C_{k,r}^{(t)}(\mathbf{w}) \iff \boldsymbol{h}_{\mathbf{v},k}^{(t)} = \boldsymbol{h}_{\mathbf{w},k}^{(t)}.$$

The following result implies that increasing $k$ leads to a strict boost in terms of expressivity of the $k$-RLWL and $k$-RN architectures in terms of distinguishing non-isomorphic multi-relational graphs.

**Proposition 8.** *For $k \geq 2$ and $r \geq 1$, there exists a pair of non-isomorphic multi-relational graphs $G_r = (V(G_r), R_1(G_r), \ldots, R_r(G_r), \ell)$ and $H = (V(H_r), R_1(H_r), \ldots, R_r(H_r), \ell)$ such that:*

- *For all choices of $\mathsf{UPD}^{(t)}$, $\mathsf{AGG}^{(t)}$, for $t > 0$, and* READOUT *the $k$-RN architecture will not distinguish the graphs $G_r$ and $H_r$.*

- *There exists $\mathsf{UPD}^{(t)}$, $\mathsf{AGG}^{(t)}$, for $t > 0$, and* READOUT *such that the $(k+1)$-RN will distinguish them.*

Moreover, the following results shows that for $k = 2$ the $k$-RN architecture is strictly more expressive than CompGCN and R-GCN in distinguishing non-isomophics graphs.

**Corollary 9.** *There exists a 2-RN architecture that is strictly more expressive than the CompGCN and the R-GCN architecture in terms of distinguishing non-isomorphic graphs.*

$k$**-RNs for Vertex-level Prediction.**    As defined in Equations 6 and 7, an $k$-RN architecture either computes $k$-tuple- or graph-level features. However, it is straightforward to compute a vertex-level features, see, e.g., Morris et al. (2022b, Section 4.1).

**Scalability.**    Although the $k$-RN is provably expressive, see Proposition 8, it suffer some high memory requirement. Similar to the $k$-WL, it's memory complexity can only be lower bounded in $\Omega(n^k)$, making it not applicable for large knowledge graphs. However, recent progress in making higher-order architectures more scalable, e.g., Morris et al. (2022b); Bevilacqua et al. (2021); Qian et al. (2022), can be straightforwardly lifted to the multi-relational case.

## 5   Experimental Study

Here, we investigate to what extend the above theoretical results hold for real-world data distributions. Specifically, we aim to answer the following questions.

**Q1**  Does the theoretical equivalence of R-GCN and CompGCN hold in practice?

**Q2**  Does the performance depend on the dimension of vertex features?

**Q3**  Does CompGCN benefit from normalization and learnable edge weights?

**Q4**  Does the theoretical difference in composition functions of CompGCN hold in practice?

**Datasets.**    To answer Q1 to Q4, we investige R-GCN and CompGCN's empirical performance on the small-scale AIFB (6 000 vertices) and the large-scale AM (1.6 million vertices) (Ristoski et al., 2016) vertex classification benchmark dataset; see Section F for dataset statistics.

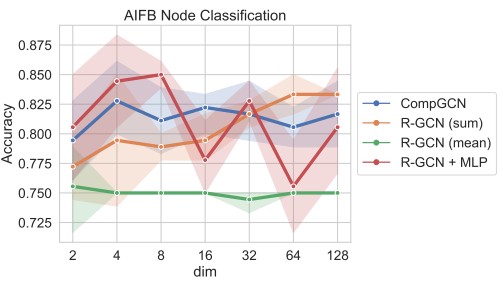
**(a)** AIFB results with varying input feature dimension.

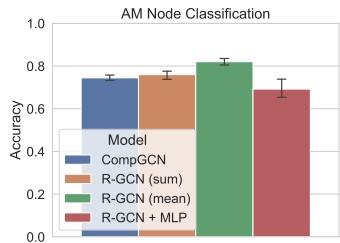
**(b)** AM results with $dim = 4$.

**Figure 1:** Vertex classification performance of CompGCN and R-GCN on smaller (AIFB) and larger (AM) graphs. Initial vertex feature dimensions higher than 4 do not improve the accuracy.

**Featurization.** Most relational GNNs for vertex- and link-level tasks assume that the initial vertex states come from a learnable vertex embedding matrix (Wang et al., 2021; Ali et al., 2021b). However, this vertex feature initialization or featurization method makes the model inherently transductive, i.e., the model must be re-trained when adding new vertices. Moreover, such an initialization strategy is incompatible with our Weisfeiler-Leman-based theoretical results since a learnable vertex embedding matrix will result in most initial vertex features being pair-wise different. Here, however, being faithful to the Weisfeiler-Leman formulation, we initialize *all* vertex features with the *same* $d$-dimensional vector, namely, a standard basis vector of $\mathbb{R}^d$, e.g., $(1, 0, \ldots, 0)$ in $\mathbb{R}^d$.[6] Relation-specific weight matrices in the case of R-GCN and edge features in the case of CompGCN are still learnable. We stress here that such a featurization strategy endows GNNs with inductive properties. Since we are using the same vertex feature initialization, we can run inference on previously unseen vertices or graphs.

**Implementation.** We use the R-GCN and CompGCN implementation provided by PyG framework (Fey and Lenssen, 2019). The source code of all methods and evaluation procedures is available at https://github.com/migalkin/RWL. For the smaller AIFB dataset, both models use two GNN layers. For the larger AM dataset, R-GCN saturates with three layers. Following the theory, we do not use any basis decomposition of relation weights in R-GCN. We list other hyperparameters in Section F. We report averaged results of five independent runs using different random seeds. We conducted all experiments in the full-batch mode on a single GPU (Tesla V100 32 GB or RTX 8000).

**Discussion.** Probing R-GCN with different aggregations and CompGCN on the smaller AIFB (Figure 1a) and larger AM (Figure 1b) datasets, we largely confirm the theoretical hypothesis of their expressiveness equivalence (**Q1**) and observe similar performance of both GNNs. The higher variance on AIFB is due to the small test set size (36 vertices), i.e., one misclassified vertex drops accuracy by $\approx 3\%$.

To test if increasing the input vertex feature dimensions leads to more expressive GNN architectures (**Q2**), we vary the initial vertex feature dimension in $\{2, 4, 8, \ldots, 64, 128\}$ on the smaller AIFB dataset (Figure 1a) and do not observe any significant differences starting from $d = 4$ and above. Having identified that, we report the best results of compared models on the larger AM graph with the vertex feature dimension $d$ in $\{4, 8\}$.

Following the theory where the sum aggregator is most expressive, we investigate this finding on the smaller AIFB dataset for both GNNs. R-GCN with mean aggregation shows slightly better results on the larger AM dataset, which we attribute to the unstable optimization process of the sum aggregator where vertices might have thousands of neighbors, leading to large losses and noisy gradients. We hypothesize that stabilizing the training process on larger graphs might improve performance.

Furthermore, we perform an ablation study (Figure 2) of main CompGCN components (**Q3**), i.e., direction-based weighting (over direct, inverse, and self-loop edges), relation projection update in each layer, and message normalization in the GCN style $\mathbf{D}^{-\frac{1}{2}} \mathbf{A} \mathbf{D}^{-\frac{1}{2}}$; see also Sections E and D.

---

[6]We also probed a vector initialized with the Glorot and Bengio (2010) strategy, showing similar results.

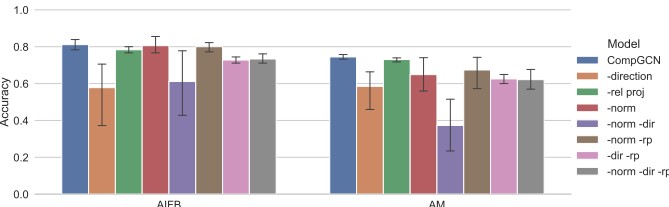

**Figure 2:** CompGCN ablations. Directionality (*-dir*) and normalization (*-norm*) are the most crucial components, i.e., their removal does lead to significant performance drops.

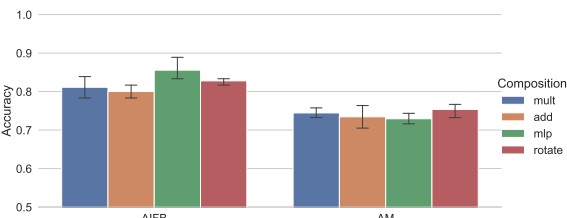

**Figure 3:** CompGCN with different composition functions. No significant differences.

The crucial components for the smaller and larger graphs are (1) three-way direction-based message passing and (2) normalization. Replacing message passing over three directions (and three weight matrices) with one weight matrix using a single adjacency leads to a significant drop in performance. Removing normalization increases variance in the larger graph. Finally, removing both directionality and normalization leads to significant degradation in predictive performance.

Studying composition functions (Figure 3), we do not find significant differences among non-parametric `mult`, `add`, `rotate` functions (**Q4**); see Section E. Performance of an MLP over a concatenation of vertex and edge features falls within confidence intervals of other compositions and does not exhibit a significant accuracy boost.

## 6   Conclusion

Here, we investigated the expressive power of two popular GNN architectures for knowledge or multi-relational graphs, namely, CompGCN and R-GCN. By deriving a variant of the 1-WL, we quantified their limits in distinguishing vertices in multi-relational graphs. Further, we investigated under which conditions, i.e., the choice of the composition function, CompGCN, reaches the same expressive power as R-GCN. To overcome the limitations of the two architectures, we derived the provably more powerful $k$-RN architecture. By increasing $k$, the $k$-RN architecture gets strictly more expressive. Empirically, we verified that our theoretical results translate largely into practice. Using CompGCN and R-GCN in a vertex classification setting over small and large multi-relational graphs shows that both architectures provide a similar performance level. We believe that our paper is the first step in a principled design of GNNs for knowledge or multi-relational graphs.

## Acknowledgements

Pablo Barceló is funded by Fondecyt grant 1200967, the ANID - Millennium Science Initiative Program - Code ICN17002, and the National Center for Artificial Intelligence CENIA FB210017, Basal ANID. Mikhail Galkin is funded by the Samsung AI grant held at Mila. Christopher Morris is partially funded a DFG Emmy Noether grant (468502433) and RWTH Junior Principal Investigator Fellowship under the Excellence Strategy of the Federal Government and the Länder. Miguel Romero is funded by Fondecyt grant 11200956, the Data Observatory Foundation, and the National Center for Artificial Intelligence CENIA FB210017, Basal ANID.

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

# A   Appendix

## A.1   Related Work

In the following, we expand on relevant related work.

**GNNs.**   Recently, GNNs (Gilmer et al., 2017; Scarselli et al., 2009) emerged as the most prominent graph representation learning architecture. Notable instances of this architecture include, e.g., Duvenaud et al. (2015); Hamilton et al. (2017) and Veličković et al. (2018), which can be subsumed under the message-passing framework introduced in Gilmer et al. (2017). In parallel, approaches based on spectral information were introduced in, e.g., Defferrard et al. (2016); Bruna et al. (2014); Kipf and Welling (2017) and Monti et al. (2017)—all of which descend from early work in Scarselli et al. (2009); Baskin et al. (1997); Kireev (1995); Micheli and Sestito (2005); Merkwirth and Lengauer (2005); Micheli (2009) and Sperduti and Starita (1997).

**Limits of GNNs and More Expressive Architectures.**   Recently, connections between GNNs and Weisfeiler–Leman type algorithms have been shown (Morris et al., 2019; Xu et al., 2019). Specifically, Morris et al. (2019) and Xu et al. (2019) showed that the 1-WL limits the expressive power of any possible GNN architecture in terms of distinguishing non-isomorphic graphs. In turn, these results have been generalized to the $k$-WL, see, e.g., Morris et al. (2019); Azizian and Lelarge (2020); Geerts et al. (2020); Geerts (2020); Maron et al. (2019); Morris et al. (2020, 2022a), and connected to permutation-equivariant function approximation over graphs, see, e.g., Chen et al. (2019); Geerts and Reutter (2022); Maehara and NT (2019). Barceló et al. (2020) further established an equivalence between the expressiveness of GNNs with readout functions and $\mathsf{C}^2$, the 2-variable fragment of first-order logic extended by counting quantifiers.

**Relational GNNs.**   Relational GNNs enjoy a profound usage in many areas of machine learning, such as complex question answering in NLP (Fu et al., 2020) or visual question answering (Huang et al., 2022) in the intersection of NLP and vision. Notably, Schlichtkrull et al. (2018) proposed the first architecture, namely, R-GCN, being able to handle multi-relational data. Further, Vashishth et al. (2020) proposed an alternative GNN architecture, namely, CompGCN, using less number of parameters and reporting improved empirical performance. In the knowledge graph reasoning area, R-GCN and CompGCN, being strong baselines, spun off numerous improved GNNs for vertex classification and transductive link prediction tasks (Galkin et al., 2020; Yu et al., 2020; Zhang et al., 2022). Furthermore, they inspired architectures for more complex reasoning tasks such as inductive link prediction (Zhu et al., 2021; Teru et al., 2020; Ali et al., 2021a; Zhang and Yao, 2022) and logical query answering (Daza and Cochez, 2020; Alivanistos et al., 2022; Zhu et al., 2022).

Despite various applications, there has not been any theoretical work shedding light on multi-relational GNNs' expressive power and learning performance. Some recent empirical results highlight interesting properties of relational GNNs, e.g., a randomly initialized and untrained R-GCN still demonstrates non-trivial performance (Degraeve et al., 2022), or that random perturbation of the relations does not lead to performance drops for CompGCN (Zhang et al., 2022).

## A.2   The Weisfeiler–Leman Algorithm

In the following, we briefly describe Weisfeiler–Leman-type algorithms, starting with the 1-*dimensional Weisfeiler–Leman algorithm* (1-WL).

**The 1-WL.**   The 1-WL, or *color refinement*, is a simple heuristic for the graph isomorphism problem, originally proposed by Weisfeiler and Leman (1968).[7] Intuitively, the algorithm determines if two graphs are non-isomorphic by iteratively coloring or labeling vertices. Given an initial coloring or labeling of the vertices of both graphs, e.g., their degree or application-specific information, in each iteration, two vertices with the same label get different labels if the number of identically labeled neighbors is not equal. If, after some iteration, the number of vertices annotated with a specific label is different in both graphs, the algorithm terminates and a stable coloring (partition) is obtained. We can then conclude that the two graphs are not isomorphic. It is easy to see that the algorithm cannot

---

[7]Strictly speaking, 1-WL and color refinement are two different algorithms. That is, 1-WL considers neighbors and non-neighbors to update the coloring, resulting in a slightly higher expressive power when distinguishing vertices in a given graph, see Grohe (2021) for details. For brevity, we consider both algorithms to be equivalent.

distinguish all non-isomorphic graphs (Cai et al., 1992a). Nonetheless, it is a powerful heuristic that can successfully test isomorphism for a broad class of graphs (Arvind et al., 2015; Babai and Kucera, 1979; Kiefer et al., 2015).

Formally, let $G = (V(G), E(G), \ell)$ be a labeled graph. In each iteration, $t > 0$, the 1-WL computes a vertex coloring $C^{(t)} \colon V(G) \to \mathbb{N}$, which depends on the coloring of the neighbors. That is, in iteration $t > 0$, we set

$$C^{(t)}(v) \coloneqq \mathsf{RELABEL}\Big(\big(C^{(t-1)}(v), \{\!\{C^{(t-1)}(u) \mid u \in N(v)\}\!\}\big)\Big),$$

where RELABEL injectively maps the above pair to a unique natural number, which has not been used in previous iterations. In iteration 0, the coloring $C^{(0)} \coloneqq \ell$. To test if two graphs $G$ and $H$ are non-isomorphic, we run the above algorithm in "parallel" on both graphs. If the two graphs have a different number of vertices colored $c$ in $\mathbb{N}$ at some iteration, the 1-WL *distinguishes* the graphs as non-isomorphic. Moreover, if the number of colors between two iterations, $t$ and $(t + 1)$, does not change, i.e., the cardinalities of the images of $C^{(t)}$ and $C^{(t+1)}$ are equal, or, equivalently,

$$C^{(t)}(v) = C^{(t)}(w) \iff C^{(t+1)}(v) = C^{(t+1)}(w),$$

for all vertices $v$ and $w$ in $V(G)$, the algorithm terminates. For such $t$, we define the *stable coloring* $C^\infty(v) = C^{(t)}(v)$ for $v$ in $V(G)$. The stable coloring is reached after at most $\max\{|V(G)|, |V(H)|\}$ iterations (Grohe, 2017).

Due to the shortcomings of the 1-WL or color refinement in distinguishing non-isomorphic graphs, several researchers, e.g., (Babai, 1979; Immerman and Lander, 1990), devised a more powerful generalization of the former, today known as the $k$-dimensional Weisfeiler-Leman algorithm ($k$-WL).[8]

**Oblivious $k$-WL.**   Intuitively, to surpass the limitations of the 1-WL, the $k$-WL colors ordered subgraphs instead of a single vertex. More precisely, given a graph $G$, it colors the tuples from $V(G)^k$ for $k \geq 2$ instead of the vertices. By defining a neighborhood between these tuples, we can define a coloring similar to the 1-WL. Formally, let $G$ be a labeled graph, and let $k \geq 2$. In each iteration $t \geq 0$, the algorithm, similarly to the 1-WL, computes a *coloring* $C_k^{(t)} \colon V(G)^k \to \mathbb{N}$. In the first iteration, $t = 0$, the tuples $\mathbf{v}$ and $\mathbf{w}$ in $V(G)^k$ get the same color if they have the same atomic type, i.e., $C_k^{(0)}(\mathbf{v}) \coloneqq \mathsf{atp}(\mathbf{v})$. Now, for $t \geq 0$, $C_k^{(t+1)}$ is defined by

$$C_{(t+1)}^k(\mathbf{v}) \coloneqq \mathsf{RELABEL}\Big(\big(C_k^{(t)}(\mathbf{v}), M^{(t)}(\mathbf{v})\big)\Big),$$

with $M^{(t)}(\mathbf{v})$ the tuple

$$M^{(t)}(\mathbf{v}) \coloneqq \big(\{\!\{C_k^{(t)}(\theta_1(\mathbf{v}, w)) \mid w \in V(G)\}\!\}, \ldots, \{\!\{C_k^{(t)}(\theta_k(\mathbf{v}, w)) \mid w \in V(G)\}\!\}\big). \qquad (8)$$

We also call $M^{(t)}$ an *aggregation function*. Here

$$\theta_j(\mathbf{v}, w) \coloneqq (v_1, \ldots, v_{j-1}, w, v_{j+1}, \ldots, v_k).$$

That is, $\theta_j(\mathbf{v}, w)$ replaces the $j$-th component of the tuple $\mathbf{v}$ with the vertex $w$. Hence, two tuples $\mathbf{v}$ and $\mathbf{w}$ with the same color in iteration $t$ get different colors in iteration $(t + 1)$ if there exists a $j$ in $[k]$ such that the number of $j$-neighbors of $\mathbf{v}$ and $\mathbf{w}$, respectively, colored with a certain color is different.

Hence, two tuples are *adjacent* or *$j$-neighbors* if they are different in the $j$th component (or equal, in the case of self-loops). Again, we run the algorithm until convergence, i.e.,

$$C_k^{(t)}(\mathbf{v}) = C_k^{(t)}(\mathbf{w}) \iff C_k^{(t+1)}(\mathbf{v}) = C_k^{(t+1)}(\mathbf{w}),$$

for all $\mathbf{v}$ and $\mathbf{w}$ in $V(G)^k$ holds, and call the partition of $V(G)^k$ induced by $C_k^{(t)}$ the stable partition. For such $t$, we define $C_k^\infty(\mathbf{v}) \coloneqq C_k^{(t)}(\mathbf{v})$ for $\mathbf{v}$ in $V(G)^k$.

---

[8]There exists two definitions of the $k$-WL, the so-called oblivious $k$-WL and the folklore or non-oblivious $k$-WL, see Grohe (2021). There is a subtle difference in how they aggregate neighborhood information. Within the graph learning community, it is customary to abbreviate the oblivious $k$-WL as $k$-WL, a convention that we follow in this paper.

To test whether two graphs $G$ and $H$ are non-isomorphic, we run the $k$-WL in "parallel" on both graphs. Then, if the two graphs have a different number of $k$-tuples colored $c$ in $\mathbb{N}$, the $k$-WL *distinguishes* the graphs as non-isomorphic. By increasing $k$, the algorithm becomes more powerful in distinguishing non-isomorphic graphs, i.e., for each $k \geq 1$, there are non-isomorphic graphs distinguished by $(k+1)$-WL but not by $k$-WL (Cai et al., 1992a).

**Local $\delta$-$k$-dimensional Weisfeiler–Leman Algorithm.** Morris et al. (2020) introduced a more efficient variant of the $k$-WL, the *local $\delta$-$k$-dimensional Weisfeiler–Leman algorithm* ($\delta$-$k$-LWL). In contrast to the $k$-WL, the $\delta$-$k$-LWL considers only a subset of the entire neighborhood of a vertex tuple. Let the tuple $\mathbf{w} = \theta_j(\mathbf{v}, w)$ be a $j$-neighbor of $\mathbf{v}$. We say that $\mathbf{w}$ is a *local $j$-neighbor* of $\mathbf{v}$ if $w$ is adjacent to the replaced vertex $v_j$. Otherwise, the tuple $\mathbf{w}$ is a *global $j$-neighbor* of $\mathbf{v}$. The $\delta$-$k$-LWL considers only local neighbors during the neighborhood aggregation process, and discards any information about the global neighbors. Formally, the $\delta$-$k$-LWL refines a coloring $C_{k,\delta}^{(t)}$ (obtained after $t$ rounds of the $\delta$-$k$-LWL) via the aggregation function

$$M_\delta^{(t)}(\mathbf{v}) := \left( \{\!\{ C_{k,\delta}^{(t)}(\theta_1(\mathbf{v}, w)) \mid w \in N(v_1) \}\!\}, \ldots, \{\!\{ C_{k,\delta}^{(t)}(\theta_k(\mathbf{v}, w)) \mid w \in N(v_k) \}\!\} \right),$$

hence considering only the local $j$-neighbors of the tuple $\mathbf{v}$ in each iteration. The coloring functions for the iterations of the $\delta$-$k$-LWL are then defined by

$$C_{k,\delta}^{(t+1)}(\mathbf{v}) := \mathsf{RELABEL}\left( \left( C_{k,\delta}^{(t)}(\mathbf{v}), M_\delta^{(t)}(\mathbf{v}) \right) \right).$$

Note that the 1-WL is equivalent to the $\delta$-1-LWL. Morris et al. (2020) showed that, for each $k$, the $\delta$-$k$-LWL can distinguish graphs that the $k$-WL cannot and derived a variation of the former that is strictly more powerful than the $k$-WL.

# B    Missing Proofs in Section 3

**Theorem 10** (Theorem 1 in the main text). *Let $G = (V(G), R_1(G), \ldots, R_r(G), \ell)$ be a labeled, multi-relational graph. Then for all $t \geq 0$ the following hold:*

- *For all choices of initial vertex features consistent with $\ell$, sequences $\mathbf{W}_{R\text{-}GCN}^{(t)}$ of R-GCN parameters, and vertices $v$ and $w$ in $V(G)$,*

$$C_R^{(t)}(v) = C_R^{(t)}(w) \implies \boldsymbol{h}_{v,R\text{-}GCN}^{(t)} = \boldsymbol{h}_{w,R\text{-}GCN}^{(t)}.$$

- *For all choices of initial vertex features consistent with $\ell$, sequences $\mathbf{W}_{CompGCN}^{(t)}$ of CompGCN parameters, composition functions $\phi$, and vertices $v$ and $w$ in $V(G)$,*

$$C_R^{(t)}(v) = C_R^{(t)}(w) \implies \boldsymbol{h}_{v,CompGCN}^{(t)} = \boldsymbol{h}_{w,CompGCN}^{(t)}.$$

*Proof.* We only prove it for CompGCN as the proof for R-GCN is analogous. Fix initial vertex features $(\boldsymbol{h}_v^{(0)})_{v \in V(G)}$ for $G$ consistent with $\ell$, a sequence $\mathbf{W}_{\mathsf{CompGCN}}^{(t)}$ of parameters, a composition function $\phi$, and two vertices $v$ and $w$ in $V(G)$. We prove the result by induction on $t \geq 0$. For $t = 0$, the statement follows immediately from the fact the initial features $(\boldsymbol{h}_v^{(0)})_{v \in V(G)}$ are consistent with $\ell$. Assume now that $C_R^{(t)}(v) = C_R^{(t)}(w)$, for $t > 0$. Hence, by Equation 4, it must be the case that

- $C_R^{(t-1)}(v) = C_R^{(t-1)}(w)$, and

- $\{\!\{ C_R^{(t-1)}(u) \mid u \in N_i(v) \}\!\} = \{\!\{ C_R^{(t-1)}(u) \mid u \in N_i(w) \}\!\}$, for each $i \in [r]$.

Then, by induction hypothesis, it holds that:

- $\boldsymbol{h}_{v,\mathsf{CompGCN}}^{(t-1)} = \boldsymbol{h}_{w,\mathsf{CompGCN}}^{(t-1)}$, and

- $\{\!\{ \boldsymbol{h}_{u,\mathsf{CompGCN}}^{(t-1)} \mid u \in N_i(v) \}\!\} = \{\!\{ \boldsymbol{h}_{u,\mathsf{CompGCN}}^{(t-1)} \mid u \in N_i(w) \}\!\}$, for each $i$ in $[r]$.

From these two we conclude by applying Equation 3 that $\boldsymbol{h}_{v,\mathsf{CompGCN}}^{(t)} = \boldsymbol{h}_{w,\mathsf{CompGCN}}^{(t)}$. This is because we have that $\boldsymbol{h}_{v,\mathsf{CompGCN}}^{(t-1)} \boldsymbol{W}_0^{(t)} = \boldsymbol{h}_{w,\mathsf{CompGCN}}^{(t-1)} \boldsymbol{W}_0^{(t)}$ and

$$\sum_{u \in N_i(v)} \phi\big(\boldsymbol{h}_{u,\mathsf{CompGCN}}^{(t-1)}, \boldsymbol{z}_i^{(t)}\big) \boldsymbol{W}_1^{(t)} = \sum_{u \in N_i(w)} \phi\big(\boldsymbol{h}_{u,\mathsf{CompGCN}}^{(t-1)}, \boldsymbol{z}_i^{(t)}\big) \boldsymbol{W}_1^{(t)},$$

for each $i \in [r]$. $\qquad\square$

**Theorem 11** (Theorem 2 in the main text)**.** *Let $G = (V(G), R_1(G), \ldots, R_r(G), \ell)$ be a labeled, multi-relational graph. For all $t \geq 0$:*

- *There exist initial vertex features and a sequence $\mathbf{W}_{R\text{-}GCN}^{(t)}$ of parameters such that for all $v$ and $w$ in $V(G)$,*
$$C_R^{(t)}(v) = C_R^{(t)}(w) \iff \boldsymbol{h}_{v,R\text{-}GCN}^{(t)} = \boldsymbol{h}_{w,R\text{-}GCN}^{(t)}.$$

- *There exist initial vertex features, a sequence $\mathbf{W}_{CompGCN}^{(t)}$ of parameters and a composition function $\phi$ such that for all $v$ and $w$ in $V(G)$,*
$$C_R^{(t)}(v) = C_R^{(t)}(w) \iff \boldsymbol{h}_{v,CompGCN}^{(t)} = \boldsymbol{h}_{w,CompGCN}^{(t)}.$$

*Proof.* We focus on the case of CompGCN when the composition map $\phi$ is vector scaling, that is, $\phi(\boldsymbol{h}, \alpha) = \alpha\boldsymbol{h}$, for $\boldsymbol{h}$ in $\mathbb{R}^d$ and $\alpha$ in $\mathbb{R}$. As we explain later, this implies the cases of R-GCN, CompGCN with point-wise multiplication, and also CompGCN with circular-correlation.

The proof is a refinement of (Morris et al., 2019, Theorem 2) for multi-relational graphs. For a matrix $\boldsymbol{B}$, we denote by $\boldsymbol{B}_i$ its $i$-th row. Let $n = |V(G)|$ and without loss of generality assume $V(G) = [n]$. We represent vertex features for $G$ as a matrix $\boldsymbol{F}$ in $\mathbb{R}^{n \times d}$, where $\boldsymbol{F}_v$ corresponds to the vertex feature of $v$. By slightly abusing notation, we view vertex features as a coloring for $G$. In particular, we denote by $\Gamma_G(\boldsymbol{F})$ the application of one step of the 1-RWL on $G$. That is, $\Gamma_G(\boldsymbol{F})$ is a coloring $C \colon V(G) \to \mathbb{N}$ such that for each $v$ in $V(G)$,

$$C(v) := \mathsf{RELABEL}\Big(\big(C_{\boldsymbol{F}}(v), \{\!\{(C_{\boldsymbol{F}}(u), i) \mid i \in [r], u \in N_i(v)\}\!\}\big)\Big),$$

where $C_{\boldsymbol{F}}$ is the coloring corresponding to the matrix $\boldsymbol{F}$. On the other hand, the update rule of CompGCN can be written as follows:

$$\boldsymbol{F}' = \sigma\big(\boldsymbol{F}\boldsymbol{W}_0 + \sum_{i \in [r]} \alpha_i \boldsymbol{A}_i \boldsymbol{F} \boldsymbol{W}_1 + b\boldsymbol{J}\big),$$

where $\boldsymbol{W}_0$ and $\boldsymbol{W}_1$ are the parameter matrices, $\alpha_i$ are the scaling factors, $\boldsymbol{A}_i$ is the adjacency matrix for the relation $R_i(G)$, and $\boldsymbol{J}$ is the all-one matrix of appropriate dimensions, representing the biases. Here we choose $\sigma$ to be the sign function sign and the bias $b$ to be $b = -1$. Using the same argument as in (Morris et al., 2019, Corollary 16), we can replace $\sigma$ by the ReLU function.

We need the following lemma shown in (Morris et al., 2019, Lemma 9).

**Lemma 12** (Morris et al. (2019))**.** *Let $\boldsymbol{B}$ in $\mathbb{N}^{s \times t}$ be a matrix such that all the rows are pairwise distinct. Then there is a matrix $\boldsymbol{X}$ in $\mathbb{R}^{t \times s}$ such that the matrix $\mathsf{sign}(\boldsymbol{B}\boldsymbol{X} - \boldsymbol{J})$ in $\{-1, 1\}^{s \times s}$ is non-singular.*

Following Morris et al. (2019), we say that a matrix is *row-independent modulo equality* if the set of all rows appearing in the matrix is linearly independent. For two colorings $C$ and $C'$ of $G$, we write $C \equiv C'$ if the colorings define the same partition on $V(G)$. The key lemma of the proof is the following:

**Lemma 13.** *Let $\boldsymbol{F}$ in $\mathbb{R}^{n \times d}$ be row-independent modulo equality. Then there are matrices $\boldsymbol{W}_0$ and $\boldsymbol{W}_1$ in $\mathbb{R}^{d \times e}$ and scaling factors $\alpha_i$ in $\mathbb{R}$, for $i$ in $[r]$, such that the matrix*

$$\boldsymbol{F}' = \mathsf{sign}\big(\boldsymbol{F}\boldsymbol{W}_0 + \sum_{i \in [r]} \alpha_i \boldsymbol{A}_i \boldsymbol{F} \boldsymbol{W}_1 - \boldsymbol{J}\big)$$

*is row-independent modulo equality and $\boldsymbol{F}' \equiv \Gamma_G(\boldsymbol{F})$.*

*Proof.* Let $q$ be the number of distinct rows in $\boldsymbol{F}$ and let $\widetilde{\boldsymbol{F}}$ in $\mathbb{R}^{q \times d}$ be the matrix whose rows are the distinct rows of $\boldsymbol{F}$ in an arbitrary but fixed order. We denote by $Q_1, \ldots, Q_q$ the associated *color classes*, that is, a vertex $v$ in $[n]$ is in $Q_j$ if and only if $\boldsymbol{F}_v = \widetilde{\boldsymbol{F}}_j$. By construction, the rows of $\widetilde{\boldsymbol{F}}$ are linearly independent, and hence there is a matrix $\boldsymbol{M}$ in $\mathbb{R}^{d \times q}$ such that $\widetilde{\boldsymbol{F}}\boldsymbol{M}$ in $\mathbb{R}^{q \times q}$ is the identity matrix. It follows that the matrix $\boldsymbol{F}\boldsymbol{M}$ in $\mathbb{R}^{n \times q}$ has entries:

$$(\boldsymbol{F}\boldsymbol{M})_{vj} = \begin{cases} 1 & \text{if } v \in Q_j \\ 0 & \text{otherwise.} \end{cases}$$

Let $\boldsymbol{D}$ in $\mathbb{N}^{n \times q(r+1)}$ be the matrix with entries:

$$\boldsymbol{D}_{vh} = \begin{cases} |N_i(v) \cap Q_j| & \text{if } h = iq + j \text{ for } i \in [r], j \in [q] \\ 1 & \text{if } h \in [q] \text{ and } v \in Q_h \\ 0 & \text{otherwise.} \end{cases}$$

So the $v$-th row of $\boldsymbol{D}$ is the concatenation of a one-hot vector encoding of the color of $v$ and a vector encoding for the multiset of the colors in $N_i(v)$, for each $i$ in $[r]$. We have

$$\Gamma_G(\boldsymbol{F}) \equiv \boldsymbol{D}$$

if we view $\boldsymbol{D}$ as a coloring of $G$. We can also see $\boldsymbol{D}$ as a block matrix $\boldsymbol{D} = [\boldsymbol{N}_0\,\boldsymbol{N}_1 \cdots \boldsymbol{N}_r]$, where $\boldsymbol{N}_0 = \boldsymbol{F}\boldsymbol{M}$ in $\mathbb{N}^{n \times q}$ and $\boldsymbol{N}_i = \boldsymbol{A}_i\boldsymbol{F}\boldsymbol{M}$ in $\mathbb{N}^{n \times q}$ for each $i$ in $[r]$. Since $0 \leq \boldsymbol{D}_{vh} \leq n-1$, for all $v$ in $[n], h$ in $[q(r+1)]$, we have

$$\boldsymbol{D} \equiv \boldsymbol{E}$$

where

$$\boldsymbol{E} = \boldsymbol{F}\boldsymbol{M} + \sum_{i \in [r]} n^i \boldsymbol{A}_i \boldsymbol{F}\boldsymbol{M}.$$

Indeed, $\boldsymbol{E}_{vj}$ is simply the $n$-base representation of the vector $(\boldsymbol{D}_{vj}, \boldsymbol{D}_{v(qj)}, \ldots, \boldsymbol{D}_{v(rqj)})$, and hence $\boldsymbol{E}_v = \boldsymbol{E}_w$ if and only if $\boldsymbol{D}_v = \boldsymbol{D}_w$.

Let $p$ be the number of distinct rows in $\boldsymbol{E}$ and let $\widetilde{\boldsymbol{E}}$ in $\mathbb{N}^{p \times q}$ be the matrix whose rows are the distinct rows of $\boldsymbol{E}$ in an arbitrary but fixed order. We can apply Lemma 12 to $\widetilde{\boldsymbol{E}}$ and obtain a matrix $\boldsymbol{X}$ in $\mathbb{R}^{q \times p}$ such that $\text{sign}(\widetilde{\boldsymbol{E}}\boldsymbol{X} - \boldsymbol{J})$ in $\mathbb{R}^{p \times p}$ is non-singular. In particular, $\text{sign}(\boldsymbol{E}\boldsymbol{X} - \boldsymbol{J})$ is row-independent modulo equality and $\text{sign}(\boldsymbol{E}\boldsymbol{X} - \boldsymbol{J}) \equiv \boldsymbol{E} \equiv \Gamma_G(\boldsymbol{F})$. Let $\boldsymbol{W}_0 = \boldsymbol{W}_1 = \boldsymbol{M}\boldsymbol{X}$ in $\mathbb{R}^{d \times p}$ and $\alpha_i = n^i$ for $i$ in $[r]$. We have

$$\boldsymbol{F}' = \text{sign}(\boldsymbol{F}\boldsymbol{W}_0 + \sum_{i \in [r]} \alpha_i \boldsymbol{A}_i \boldsymbol{F}\boldsymbol{W}_1 - \boldsymbol{J})$$
$$= \text{sign}(\boldsymbol{F}\boldsymbol{M}\boldsymbol{X} + \sum_{i \in [r]} \alpha_i \boldsymbol{A}_i \boldsymbol{F}\boldsymbol{M}\boldsymbol{X} - \boldsymbol{J})$$
$$= \text{sign}(\boldsymbol{E}\boldsymbol{X} - \boldsymbol{J}).$$

Hence $\boldsymbol{F}'$ is row-independent modulo equality and $\boldsymbol{F}' = \text{sign}(\boldsymbol{E}\boldsymbol{X} - \boldsymbol{J}) \equiv \Gamma_G(\boldsymbol{F})$. $\qquad\square$

Now the theorem follows directly from Lemma 13. We start with initial vertex features $(\boldsymbol{h}_v^{(0)})_{v \in V(G)}$ consistent with $\ell$ such that different features are linearly independent. Hence the matrix $\boldsymbol{F}^{(0)}$ representing the initial features is row-independent modulo equality and we can apply iteratively Lemma 13 to obtain the required sequence $\boldsymbol{W}_{\text{CompGCN}}^{(t)}$ such that $C_{\text{R}}^{(t)} \equiv \boldsymbol{F}^{(t)}$, where $\boldsymbol{F}^{(t)}$ is the matrix representing the vertex features $(\boldsymbol{h}_{v,\text{CompGCN}}^{(t)})_{v \in V(G)}$. In particular, $C_{\text{R}}^{(t)}(v) = C_{\text{R}}^{(t)}(w) \Leftrightarrow \boldsymbol{h}_{v,\text{CompGCN}}^{(t)} = \boldsymbol{h}_{w,\text{CompGCN}}^{(t)}$, for all $v$ and $w$ in $V(G)$.

**Remark 14.** Note that the dimensions $d \times e$ of the parameter matrices at layer $t$ correspond to the number of distinct colors before ($q$) and after ($p$) the application of the layer.

The case of CompGCN with point-wise multiplication holds since we can simulate vector scaling as $\alpha \boldsymbol{h} = \boldsymbol{h} * (\alpha, \ldots, \alpha)$, where $*$ denotes point-wise multiplication. Similarly, the case of R-GCN follows as we can simulate vector scaling by setting $\boldsymbol{W}_i = \alpha_i \boldsymbol{W}_1$, for each $i$ in $[r]$.

Finally, we show that the result also holds for CompGCN with circular correlation. This composition map is defined as follows[9]:

$$(\boldsymbol{h} \star \boldsymbol{z})_i = \sum_{j=1}^{d} \boldsymbol{h}_j \boldsymbol{z}_{((i+j-2) \bmod d)+1},$$

where $\boldsymbol{h}, \boldsymbol{z}$ in $\mathbb{R}^d$, $\boldsymbol{h} \star \boldsymbol{z}$ in $\mathbb{R}^d$ and $i$ in $[d]$. We can easily simulate one layer of CompGCN with vector scaling using two layers of CompGCN with circular-correlation. Indeed, for a layer of the form

$$\boldsymbol{h}_v = \sigma\Big(\boldsymbol{g}_v \boldsymbol{W}_0 + \sum_{i\in[r]} \sum_{w\in N_i(v)} \alpha_i \boldsymbol{g}_w \boldsymbol{W}_1 + \boldsymbol{b}\Big),$$

where $\boldsymbol{g}_u$ in $\mathbb{R}^d$, for all $u$ in $V(G)$, we first use a layer of the form

$$\widetilde{\boldsymbol{h}}_v = \boldsymbol{g}_v \boldsymbol{P},$$

where $\boldsymbol{P}$ in $\mathbb{R}^{d\times d}$ reverts the vertex features, that is, all the entries are zero except for $\boldsymbol{P}_{(n-i+1)i} = 1$ for all $i$ in $[d]$, followed by a layer

$$\boldsymbol{h}_v = \sigma\Big(\widetilde{\boldsymbol{h}}_v \boldsymbol{P} \boldsymbol{W}_0 + \sum_{i\in[r]} \sum_{w\in N_i(v)} (\widetilde{\boldsymbol{h}}_v \star (0,\ldots,0,\alpha_i)) \boldsymbol{W}_1 + \boldsymbol{b}\Big).$$

$\square$

### B.1 On the Choice of the Composition Function for R-GCN Architectures

**Proposition 15** (Proposition 3 in the main text). *There exist a labeled, multi-relational graph* $G = (V(G), R_1(G), R_2(G), \ell)$ *and two vertices $v$ and $w$ in $V(G)$, such that $C_R^{(1)}(v) \neq C_R^{(1)}(w)$ but $C_{WR}^{\infty}(v) = C_{WR}^{\infty}(w)$.*

*Proof.* We have $V(G) = \{v, w, u_1, u_2\}$, $R_1(G) = \{(v, u_1), (w, u_2)\}$, $R_2(G) = \{(v, u_2), (w, u_1)\}$, $\ell(v) = \ell(w) = 0$, $\ell(u_1) = 1$ and $\ell(u_2) = 2$. Hence,

$$C_R^{(1)}(v) = \mathsf{RELABEL}\Big((0, \{\!\{(1,1),(2,2)\}\!\})\Big) \qquad C_R^{(1)}(w) = \mathsf{RELABEL}\Big((0, \{\!\{(2,1),(1,2)\}\!\})\Big),$$

that is, $C_R^{(1)}(v) \neq C_R^{(1)}(w)$. On the other hand,

$$C_{WR}^{(1)}(v) = \mathsf{RELABEL}\Big((0, \{\!\{1,2\}\!\}, 1, 1)\Big) \qquad C_{WR}^{(1)}(w) = \mathsf{RELABEL}\Big((0, \{\!\{1,2\}\!\}, 1, 1)\Big)$$

and then $C_{WR}^{\infty}(v) = C_{WR}^{\infty}(w)$. $\square$

As shown next, the expressive power of CompGCN architectures that use point-wise summation/substraction or vector concatenation is captured by this weaker form of multi-relational 1-WL.

**Theorem 16** (Theorem 4 in the main text). *Let $G = (V(G), R_1(G), \ldots, R_r(G), \ell)$ be a labeled, multi-relational graph. Then:*

- *For all $t \geq 0$, choices of initial vertex features consistent with $\ell$, sequence $\mathbf{W}_{CompGCN}^{(t)}$ of CompGCN parameters, and vertices $v$ and $w$ in $V(G)$,*

$$C_{WR}^{(t)}(v) = C_{WR}^{(t)}(w) \implies \boldsymbol{h}_{v,CompGCN}^{(t)} = \boldsymbol{h}_{w,CompGCN}^{(t)},$$

  *for either point-wise summation/substraction or concatenation as the composition map.*

- *For all $t \geq 0$, there exist initial vertex features and a sequence $\mathbf{W}_{CompGCN}^{(t)}$ of CompGCN parameters, such that for all vertices $v$ and $w$ in $V(G)$,*

$$C_{WR}^{(t)}(v) = C_{WR}^{(t)}(w) \iff \boldsymbol{h}_{v,CompGCN}^{(t)} = \boldsymbol{h}_{w,CompGCN}^{(t)},$$

  *for either point-wise summation/substraction or concatenation as the composition map.*

---

[9]For 0-indexed vectors, this is simply $(\boldsymbol{h} \star \boldsymbol{z})_i = \sum_{j=0}^{d-1} \boldsymbol{h}_j \boldsymbol{z}_{(i+j) \bmod d}$ for $0 \leq i \leq d-1$.

*Proof.* We start with the first item. We focus first on the case of CompGCN with vector concatenation. Note that if $\boldsymbol{h}$ in $\mathbb{R}^d$, $\boldsymbol{z}$ in $\mathbb{R}^b$ and $\boldsymbol{W}$ in $\mathbb{R}^{(d+b)\times e}$, then we have

$$(\boldsymbol{h}, \boldsymbol{z})\boldsymbol{W} = \boldsymbol{h}\boldsymbol{X} + \boldsymbol{z}\boldsymbol{Y},$$

where $\boldsymbol{X}$ in $\mathbb{R}^{d\times e}$ is the matrix given by the first $d$ rows of $\boldsymbol{W}$, while $\boldsymbol{Y}$ in $\mathbb{R}^{b\times e}$ is the matrix given by the last $b$ rows of $\boldsymbol{W}$. In particular, we can write

$$\begin{aligned}
\boldsymbol{h}_{v,\mathsf{CompGCN}}^{(t)} &= \sigma\Big(\boldsymbol{h}_{v,\mathsf{CompGCN}}^{(t-1)}\boldsymbol{W}_0^{(t)} + \sum_{i\in[r]}\sum_{u\in N_i(v)}(\boldsymbol{h}_{u,\mathsf{CompGCN}}^{(t-1)}, \boldsymbol{z}_i^{(t)})\boldsymbol{W}_1^{(t)}\Big) \\
&= \sigma\Big(\boldsymbol{h}_{v,\mathsf{CompGCN}}^{(t-1)}\boldsymbol{W}_0^{(t)} + \sum_{i\in[r]}\sum_{u\in N_i(v)}\boldsymbol{h}_{u,\mathsf{CompGCN}}^{(t-1)}\boldsymbol{X}_1^{(t)} + \boldsymbol{z}_i^{(t)}\boldsymbol{Y}_1^{(t)}\Big) \\
&= \sigma\Big(\boldsymbol{h}_{v,\mathsf{CompGCN}}^{(t-1)}\boldsymbol{W}_0^{(t)} + \sum_{i\in[r]}\sum_{u\in N_i(v)}\boldsymbol{h}_{u,\mathsf{CompGCN}}^{(t-1)}\boldsymbol{X}_1^{(t)} + \sum_{i\in[r]}|N_i(v)|\boldsymbol{z}_i^{(t)}\boldsymbol{Y}_1^{(t)}\Big).
\end{aligned}$$

Fix initial vertex features $(\boldsymbol{h}_v^{(0)})_{v\in V(G)}$ for $G$ consistent with $\ell$, a sequence $\boldsymbol{W}_{\mathsf{CompGCN}}^{(t)}$ of parameters and two vertices $v$ and $w$ in $V(G)$. We proceed by induction on $t \geq 0$. For $t = 0$ we are done as the features $(\boldsymbol{h}_v^{(0)})_{v\in V(G)}$ are consistent with $\ell$. Assume now that $C_{\mathsf{WR}}^{(t)}(v) = C_{\mathsf{WR}}^{(t)}(w)$, for $t > 0$. Then, by Equation 3, we have that

- $C_{\mathsf{WR}}^{(t-1)}(v) = C_{\mathsf{WR}}^{(t-1)}(w)$,

- $\{\!\{C_{\mathsf{WR}}^{(t-1)}(u) \mid i \in [r], u \in N_i(v)\}\!\} = \{\!\{C_{\mathsf{WR}}^{(t-1)}(u) \mid i \in [r], u \in N_i(w)\}\!\}$,

- $|N_i(v)| = |N_i(w)|$ for each $i \in [r]$.

Then, by induction hypothesis, it holds that:

- $\boldsymbol{h}_{v,\mathsf{CompGCN}}^{(t-1)} = \boldsymbol{h}_{w,\mathsf{CompGCN}}^{(t-1)}$, and

- $\{\!\{\boldsymbol{h}_{u,\mathsf{CompGCN}}^{(t-1)} \mid i \in [r], u \in N_i(v)\}\!\} = \{\!\{\boldsymbol{h}_{u,\mathsf{CompGCN}}^{(t-1)} \mid i \in [r], u \in N_i(w)\}\!\}$.

Then we have

- $\sum_{i\in[r]}|N_i(v)|\boldsymbol{z}_i^{(t)} = \sum_{i\in[r]}|N_i(w)|\boldsymbol{z}_i^{(t)}$, and

- $\sum_{i\in[r]}\sum_{u\in N_i(v)}\boldsymbol{h}_{u,\mathsf{CompGCN}}^{(t-1)} = \sum_{i\in[r]}\sum_{u\in N_i(w)}\boldsymbol{h}_{u,\mathsf{CompGCN}}^{(t-1)}$.

We conclude that $\boldsymbol{h}_{v,\mathsf{CompGCN}}^{(t)} = \boldsymbol{h}_{w,\mathsf{CompGCN}}^{(t)}$.

Note that the update rule for the case of point-wise summation/substraction is the same except that now $\boldsymbol{X}_1^{(t)} = \boldsymbol{Y}_1^{(t)}$. Hence exactly the same argument applies.

We now turn to the second item. We follow the same strategy and terminology as in the proof of Theorem 11. In this case, given a vertex feature matrix $\boldsymbol{F}$ in $\mathbb{R}^{n\times d}$, we denote by $\hat{\Gamma}_G(\boldsymbol{F})$ the application of one step of the weak 1-RWL. Hence, $\hat{\Gamma}_G(\boldsymbol{F})$ is a coloring $C\colon V(G) \to \mathbb{N}$ such that for each $v$ in $V(G)$,

$$C(v) = \mathsf{RELABEL}\Big(\big(C_{\boldsymbol{F}}(v), \{\!\{C_{\boldsymbol{F}}(u) \mid i \in [r], u \in N_i(v)\}\!\}, |N_1(v)|, \ldots, |N_r(v)|\big)\Big),$$

where $C_{\boldsymbol{F}}$ is the coloring corresponding to the matrix $\boldsymbol{F}$. In this case, the update rule for CompGCN with vector concatenation can be written as follows:

$$\boldsymbol{F}' = \sigma(\boldsymbol{F}\boldsymbol{W}_0 + \sum_{i\in[r]}\boldsymbol{A}_i\boldsymbol{F}\boldsymbol{X}_1 + \sum_{i\in[r]}\boldsymbol{A}_i\boldsymbol{Z}_i\boldsymbol{Y}_1 + b\boldsymbol{J}),$$

where $\boldsymbol{W}_0$ in $\mathbb{R}^{d\times e}$ and $\boldsymbol{W}_1 = \begin{bmatrix}\boldsymbol{X}_1 \\ \boldsymbol{Y}_1\end{bmatrix} \in \mathbb{R}^{(d+b)\times e}$, for $\boldsymbol{X}_1 \in \mathbb{R}^{d\times e}$, $\boldsymbol{Y}_1 \in \mathbb{R}^{b\times e}$, are the parameter matrices, $\boldsymbol{Z}_i \in \mathbb{R}^{n\times b}$ is the matrix where each row is a copy of the edge feature $\boldsymbol{z}_i \in \mathbb{R}^b$ associated with the relation $R_i(G)$, $\boldsymbol{A}_i$ is the adjacency matrix for the relation $R_i(G)$, and $\boldsymbol{J}$ is the all-one matrix of appropriate dimensions. We have the following:

**Lemma 17.** *Let $F$ in $\mathbb{R}^{n \times d}$ be row-independent modulo equality. Then there are matrices $W_0$ in $\mathbb{R}^{d \times e}$, $X_1$ in $\mathbb{R}^{d \times e}$, $Y_1$ in $\mathbb{R}^{b \times e}$ and vectors $z_i$ in $\mathbb{R}^b$, for $i$ in $[r]$ such that the matrix*

$$F' = \mathrm{sign}(FW_0 + \sum_{i \in [r]} A_i F X_1 + \sum_{i \in [r]} A_i Z_i Y_1 - J)$$

*is row-independent modulo equality and $F' \equiv \hat{\Gamma}_G(F)$.*

*Proof.* Let $q$ be the number of distinct rows in $F$ and let $\widetilde{F}$ in $\mathbb{R}^{q \times d}$ be the matrix whose rows are the distinct rows of $F$ in an arbitrary but fixed order. We denote by $Q_1, \ldots, Q_q$ the associated *color classes*, that is, a vertex $v$ in $[n]$ is in $Q_j$ if and only if $F_v = \widetilde{F}_j$. By construction, the rows of $\widetilde{F}$ are linearly independent, and hence there is a matrix $M$ in $\mathbb{R}^{d \times q}$ such that $\widetilde{F}M$ in $\mathbb{R}^{q \times q}$ is the identity matrix. It follows that the matrix $FM$ in $\mathbb{R}^{n \times q}$ has entries:

$$(FM)_{vj} = \begin{cases} 1 & \text{if } v \in Q_j \\ 0 & \text{otherwise.} \end{cases}$$

Let $M_0, M_1$ in $\mathbb{N}^{d \times (2q+r)}$, $M_2$ in $\mathbb{N}^{r \times (2q+r)}$ be the block matrices $M_0 = [M\, O\, O']$, $M_1 = [O\, M\, O']$ and $M_2 = [O''\, O''\, I]$, where $O$ in $\mathbb{R}^{d \times q}$, $O'$ in $\mathbb{R}^{d \times r}$, $O''$ in $\mathbb{R}^{r \times q}$ are all-0 matrices, and $I$ in $\mathbb{R}^{r \times r}$ is the identity matrix. For each $i$ in $[r]$, the required $z_i$ in $\mathbb{R}^r$ is the vector with all entries 0 except for the $i$-th position which is 1. Let $Z_i$ be the corresponding matrix whose rows are copies of $z_i$. We define $D$ in $\mathbb{N}^{n \times (2q+r)}$ as:

$$D = FM_0 + \sum_{i \in [r]} A_i F M_1 + \sum_{i \in [r]} A_i Z_i M_2$$

$$= \begin{bmatrix} FM & \sum_{i \in [r]} A_i FM & \sum_{i \in [r]} A_i Z_i \end{bmatrix}.$$

The $v$-th row of $FM$ encodes the color of $v$, the $v$-th row of $\sum_{i \in [r]} A_i FM$ encodes the multiset of the colors of $u$, when we range over $i$ in $[r]$ and $u$ in $N_i(v)$, and the $v$-th row of $\sum_{i \in [r]} A_i Z_i$ contains the sizes of $N_i(v)$ for all $i$ in $[r]$. Hence,

$$\hat{\Gamma}_G(F) \equiv D$$

if we view $D$ as a coloring of $G$.

Let $p$ be the number of distinct rows in $D$ and let $\widetilde{D}$ in $\mathbb{N}^{p \times (2q+r)}$ be the matrix whose rows are the distinct rows of $D$ in an arbitrary but fixed order. We apply Lemma 12 to $\widetilde{D}$ and obtain a matrix $X$ in $\mathbb{R}^{(2q+r) \times p}$ such that $\mathrm{sign}(\widetilde{D}X - J)$ in $\mathbb{R}^{p \times p}$ is non-singular. In particular, $\mathrm{sign}(DX - J)$ is row-independent modulo equality and $\mathrm{sign}(DX - J) \equiv D \equiv \hat{\Gamma}_G(F)$. Let $W_0 = M_0 X$ in $\mathbb{R}^{d \times p}$, $X_1 = M_1 X$ in $\mathbb{R}^{d \times p}$, and $Y_1 = M_2 X$ in $\mathbb{R}^{r \times p}$. We have

$$F' = \mathrm{sign}(FW_0 + \sum_{i \in [r]} A_i F X_1 + \sum_{i \in [r]} A_i Z_i Y_1 - J)$$

$$= \mathrm{sign}(FM_0 X + \sum_{i \in [r]} A_i F M_1 X + \sum_{i \in [r]} A_i Z_i M_2 X - J)$$

$$= \mathrm{sign}(DX - J).$$

Hence $F'$ is row-independent modulo equality and $F' = \mathrm{sign}(DX - J) \equiv \hat{\Gamma}_G(F)$. $\qquad \square$

The theorem follows directly by iteratively applying Lemma 17 starting with vertex features $(h_v^{(0)})_{v \in V(G)}$ consistent with $\ell$ such that different features are linearly independent.

The case of CompGCN with point-wise summation/substraction follows from the fact that this architecture can simulate CompGCN with vector concatenation. Indeed, we can simulate one layer of CompGCN with vector concatenation using two layers of CompGCN with point-wise summation/substraction. Take a layer of the form

$$h_v = \sigma\Big(g_v W_0 + \sum_{i \in [r]} \sum_{w \in N_i(v)} (g_w, z_i) W_1 + b\Big),$$

where $g_u$ in $\mathbb{R}^d$, for $u$ in $V(G)$, $W_0 \in \mathbb{R}^{d \times e}$, $W_1 \in \mathbb{R}^{(d+b) \times e}$ and $z_i \in \mathbb{R}^b$. We first use a layer

$$\widetilde{h}_v = g_v B,$$

where $B \in \mathbb{R}^{d \times (d+b)}$ is the $d \times d$ identity matrix with $b$ additional all-0 columns. So $\widetilde{h}_v = (g_v, 0, \ldots, 0) \in \mathbb{R}^{d+b}$. Then we apply a layer

$$h_v = \sigma\Big(\widetilde{h}_v W'_0 + \sum_{i \in [r]} \sum_{w \in N_i(v)} (\widetilde{h}_v + z'_i) W_1 + b\Big),$$

where $W'_0 \in \mathbb{R}^{(d+b) \times e}$ is the matrix $W_0 \in \mathbb{R}^{d \times e}$ with $b$ additional all-0 rows, while $z'_i = (0, \ldots, 0, z_i) \in \mathbb{R}^{d+b}$. □

Together with Proposition 15 and Theorem 11, this result states that CompGCN architectures based on vector summation or concatenation are provably weaker in terms of their capacity to distinguish vertices in graphs than the ones that use vector scaling.

## B.2   A comparison between R-GCN and CompGCN architectures

We proved that R-GCN and CompGCN with point-wise multiplication have the same power discriminating vertices in (multi-relational) graphs. Here we show that these architectures actually define the same functions on multi-relational graphs.

**Theorem 18.** *The following statements hold:*

- *For any sequence of parameters $\mathbf{W}^{(t)}_{\mathsf{CompGCN}}$ for CompGCN with point-wise multiplication, there is a sequence of parameters $\mathbf{W}^{(t)}_{\mathsf{R\text{-}GCN}}$ for R-GCN such that for each labeled, multi-relational graph $G = (V(G), R_1(G), \ldots, R_r(G), \ell)$ and choice of initial vertex features, we have $h^{(t)}_{v,\mathsf{R\text{-}GCN}} = h^{(t)}_{v,\mathsf{CompGCN}}$, for each $v$ in $V(G)$.*

- *Conversely, for any sequence of parameters $\mathbf{W}^{(t)}_{\mathsf{R\text{-}GCN}}$ for R-GCN, there exists a sequence of parameters $\mathbf{W}^{(2t)}_{\mathsf{R\text{-}GCN}}$ for CompGCN with point-wise multiplication such that for each labeled, multi-relational graph $G = (V(G), R_1(G), \ldots, R_r(G), \ell)$ and choice of initial vertex features, we have $h^{(2t)}_{v,\mathsf{CompGCN}} = h^{(t)}_{v,\mathsf{R\text{-}GCN}}$, for each $v$ in $V(G)$.*

*Proof.* The first item follows since we can simulate one layer of CompGCN with point-wise multiplication using one layer of R-GCN. Indeed, take a layer of the form

$$h_v = \sigma\Big(g_v W_0 + \sum_{i \in [r]} \sum_{w \in N_i(v)} (g_w * z_i) W_1\Big),$$

where $g_u, z_i \in \mathbb{R}^d$. This is equivalent to

$$h_v = \sigma\Big(g_v W_0 + \sum_{i \in [r]} \sum_{w \in N_i(v)} g_w W_i\Big),$$

where $W_i = \Lambda_i W_1$, where $\Lambda_i \in \mathbb{R}^{d \times d}$ is the diagonal matrix whose diagonal is precisely $z_i$.

For the second item, we can simulate one layer of R-GCN with two layers of CompGCN with point-wise multiplication. Take a layer

$$h_v = \sigma\Big(g_v W_0 + \sum_{i \in [r]} \sum_{w \in N_i(v)} g_w W_i\Big),$$

where $g_u \in \mathbb{R}^d$, $W_0 \in \mathbb{R}^{d \times e}$, $W_i \in \mathbb{R}^{d \times e}$. We first apply a layer

$$\widetilde{h}_v = g_v B$$

where $B \in \mathbb{R}^{d \times dr}$ is the concatenation of $r$ copies of the $d \times d$ identity matrix. In particular, $\widetilde{h}_v \in \mathbb{R}^{dr}$ is the vector $g_v$ repeated $r$ times. Then we use the layer

$$h_v = \sigma\Big(\widetilde{h}_v W'_0 + \sum_{i \in [r]} \sum_{w \in N_i(v)} (\widetilde{h}_w * z_i) W'_1\Big),$$

where $\boldsymbol{W}_0' \in \mathbb{R}^{dr \times e}$ is the matrix $\boldsymbol{W}_0 \in \mathbb{R}^{d \times e}$ with $d(r-1)$ additional all-0 rows, $\boldsymbol{W}_1' \in \mathbb{R}^{dr \times e}$ is the (vertical) concatenation of the matrices $\boldsymbol{W}_i$ for $i \in [r]$, and $\boldsymbol{z}_i \in \mathbb{R}^{dr}$ is the vector with all entries 0 except for the $d$ positions $(i-1)d + 1, \ldots, (i-1)d + d$ which contain the value 1. $\qquad\square$

**Remark 19.** A similar result holds for the case of CompGCN with point-wise summation/subtraction and CompGCN with vector concatenation. The simulations between these two architectures are implicitly given in the proof of Theorem 16.

**Remark 20.** Note that, as a consequence of Theorem 11, Proposition 15 and the first item of Theorem 16, there are functions defined by R-GCN or CompGCN with point-wise multiplication that cannot be expressed by CompGCN with point-wise summation/subtraction or vector concatenation. This even holds in the non-uniform sense, that is, if we focus on a single labeled multi-relational graph (the one from Proposition 15).

## C  Missing proofs in Section 4

**Proposition 21** (Proposition 5 in the main text). *For all $r \geq 1$, there exists a pair of non-isomorphic graphs $G = (V(G), R_1(G), \ldots, R_r(G), \ell)$ and $H = (V(H), R_1(H), \ldots, R_r(H), \ell)$ that cannot be distinguished by R-GCN or CompGCN.*

*Proof.* We explicitly construct the graphs $G$ and $H$ for $r \geq 2$. To do so, we take a pair of graphs $A$ and $B$, non-distinguishable by 1-WL, and transform them into the multi-relational graphs $G$ and $H$. Let $A$ be a cycle on six vertices and $B$ be the disjoint union of two cycles on three vertices. Clearly, the 1-WL cannot distinguish the two graphs. Now let $V(G) := V(A)$ and $V(H) := V(B)$. Further, let $R_i(G) := E(A)$ and $R_i(H) := E(B)$ for $i$ in $[r]$. Observe that the multi-relational 1-WL will reach the stable coloring after one iteration and it will not distinguish the multi-relational graphs $G$ and $H$. Hence, by Theorem 10, the result follows. $\qquad\square$

**Proposition 22** (Theorem 6 in the main text). *Let $G = (V(G), R_1(G), \ldots, R_r(G), \ell)$ be a labeled, multi-relational graph. Then for all $t \geq 0$, $r > 0$, $k \geq 1$, and all choices of $\mathsf{UPD}^{(t)}$, $\mathsf{AGG}^{(t)}$, and all $\mathbf{v}$ and $\mathbf{w}$ in $V(G)$,*
$$C_{k,r}^{(t)}(\mathbf{v}) = C_{k,r}^{(t)}(\mathbf{w}) \implies \boldsymbol{h}_{\mathbf{v},k}^{(t)} = \boldsymbol{h}_{\mathbf{w},k}^{(t)}.$$

*Proof sketch.* The proof is analogous to the proof of Morris et al. (2019, Proposition 3). $\qquad\square$

**Proposition 23** (Theorem 7 in the main text). *Let $G = (V(G), R_1(G), \ldots, R_r(G), \ell)$ be a labeled, multi-relational graph. Then for all $t \geq 0$ and $k \geq 1$, there exists $\mathsf{UPD}^{(t)}$, $\mathsf{AGG}^{(t)}$, such that for all $\mathbf{v}$ and $\mathbf{w}$ in $V(G)$,*
$$C_{k,r}^{(t)}(\mathbf{v}) = C_{k,r}^{(t)}(\mathbf{w}) \iff \boldsymbol{h}_{\mathbf{v},k}^{(t)} = \boldsymbol{h}_{\mathbf{w},k}^{(t)}.$$

*Proof.* To prove the results, we need to ensure that there exists instantiations of $\mathsf{UPD}^{(t)}$ and $\mathsf{AGG}^{(t)}$ that are injective. To show the existence of injective instantiations of $\mathsf{AGG}^{(t)}$ for $t > 0$, we write $\mathsf{AGG}^{(t)}$ as

$$\mathsf{AGG}_{\text{out}}^{(t)}\Big(\mathsf{AGG}_{\text{in},1}^{(t)}\big(\{\!\!\{\phi(\boldsymbol{h}_{\theta_1(\mathbf{v},w),k}^{(t-1)}, \boldsymbol{z}_i^{(t)}) \mid w \in N_i(v_1) \text{ and } i \in [r]\}\!\!\}\big), \ldots,$$
$$\mathsf{AGG}_{\text{in},k}^{(t)}\big(\{\!\!\{\phi(\boldsymbol{h}_{\theta_k(\mathbf{v},w),k}^{(t-1)}, \boldsymbol{z}_i^{(t)}) \mid w \in N_i(v_k) \text{ and } i \in [r]\}\!\!\}\big)\Big),$$

where $\mathsf{AGG}_{\text{out}}^{(t)}$ and $\mathsf{AGG}_{\text{in},j}^{(t)}$ for $j$ in $[k]$ may be a differentiable parameterized functions, e.g., neural networks. Observe that we can represent $\mathsf{AGG}_{\text{in},j}^{(t)}$ as

$$\sum_{i \in [r]} \sum_{w \in N_i(v_j)} \phi(\boldsymbol{h}_{\theta_j(\mathbf{v},w)}^{(t-1)}, \boldsymbol{z}_i^{(t)}) \cdot \boldsymbol{W}_1^{(t)},$$

for $j$ in $[k]$, resembling the aggregation of Equation 3, by Theorem 11, the injectiveness of the above aggregation function follows. A similar argument can be made for $\mathsf{AGG}_{\text{out}}^{(t)}$ and $\mathsf{UPD}^{(t)}$, implying the result. $\qquad\square$

Moreover, the following result implies that increasing $k$ leads to a strict boost in terms of expressivity of the $k$-RLWL and $k$-RNs architectures in terms of distinguishing non-isomorphic multi-relational graphs.

**Proposition 24.** *For $k \geq 2$ and $r \geq 1$, there exists a pair of non-isomorphic multi-relational graphs $G_r = (V(G_r), R_1(G_r), \ldots, R_r(G_r), \ell)$ and $H_r = (V(H_r), R_1(H_r), \ldots, R_r(H_r), \ell)$ that can be distinguished by the $(k+1)$-MLWL but not by the $k$-MLWL.*

*Proof.* See Proof C.1. $\square$

**Corollary 25** (Corollary 8 in the main text). *For $k \geq 2$ and $r \geq 1$, there exists a pair of non-isomorphic multi-relational graphs $G_r = (V(G_r), R_1(G_r), \ldots, R_r(G_r), \ell)$ and $H = (V(H_r), R_1(H_r), \ldots, R_r(H_r), \ell)$ such that:*

- *For all choices of $\mathsf{UPD}^{(t)}$, $\mathsf{AGG}^{(t)}$, for $t > 0$, and $\mathsf{READOUT}$ the $k$-RN architecture will not distinguish the graphs $G_r$ and $H_r$.*

- *There exists $\mathsf{UPD}^{(t)}$, $\mathsf{AGG}^{(t)}$, for $t > 0$, and $\mathsf{READOUT}$ such that the $(k+1)$-RN will distinguish them.*

*Proof.* Follows from Theorem 23 and Theorem 24. $\square$

**Corollary 26.** *There exists a 2-RN architecture that is strictly more expressive than the CompGCN and the R-GCN architecture in terms of distinguishing non-isomorphic graphs.*

*Proof.* This follows from Corollary 25 and the fact that a 2-RN is capable to distinguish the graphs constructed in the proof of Proposition 21, which follows from the fact that the $\delta$-2-LWL can distinguish the graphs $A$ and $B$; see, e.g., the proof of Lemma 13 in Morris et al. (2022b). $\square$

## C.1 Proof of Proposition 24

In the following, we outline the proof of Theorem 24. We modify the construction employed in (Morris et al., 2020), Appendix C.1.1., where they provide an infinite family of graphs $(G_k, H_k)_{k \in \mathbb{N}}$ such that the $k$-WL does not distinguish $G_k$ and $H_k$, although the $\delta$-$k$-LWL distinguishes $G_k$ and $H_k$. We recall some relevant definitions from their paper.

**Construction of $G_k$ and $H_k$.** Let $K$ denote the complete graph on $k + 1$ vertices (without any self-loops). The vertices of $K$ are indexed from 0 to $k$. Let $E(v)$ denote the set of edges incident to $v$ in $K$: clearly, $|E(v)| = k$ for all $v$ in $V(K)$. We call the elements of $V(K)$ *base vertices*, and the elements of $E(K)$ *base edges*. Define the graph $G_k$ as follows:

1. For the vertex set $V(G_k)$, we add
    (a) $(v, S)$ for each $v$ in $V(K)$ and for each *even* subset $S$ of $E(v)$,
    (b) two vertices $e^1, e^0$ for each edge $e$ in $E(K)$.
2. For the edge set $E(G_k)$, we add
    (a) an edge $\{e^0, e^1\}$ for each $e$ in $E(K)$,
    (b) an edge between $(v, S)$ and $e^1$ if $v$ in $e$ and $e$ in $S$,
    (c) an edge between $(v, S)$ and $e^0$ if $v$ in $e$ and $e$ not in $S$,

Define a companion graph $H_k$, in a similar manner to $G_k$, with the following exception: in Step 1(a), for the vertex 0 in $V(K)$, we choose all *odd* subsets of $E(0)$.

*Distance-two-cliques.* A set $S$ of vertices is said to form a *distance-two-clique* if the distance between any two vertices in $S$ is exactly 2. The following results were shown in (Morris et al., 2020).

**Lemma 27** ((Morris et al., 2020)). *The following holds for the graphs $G_k$ and $H_k$ defined above.*

- *There exists a distance-two-clique of size $(k + 1)$ inside $G_k$.*

- *There does not exist a distance-two-clique of size $(k + 1)$ inside $H_k$.*

Hence, $G_k$ and $H_k$ are non-isomorphic.

**Lemma 28** (([Morris et al., 2020])). *The $\delta$-$k$-LWL distinguishes $G_k$ and $H_k$, while the (oblivious) $k$-WL does not distinguish $G_k$ and $H_k$.*

Moreover, we need the following result showing that the $\delta$-$k$-LWL forms a hierarchy.

**Lemma 29.** *For $k \geq 2$, the $\delta$-$k$-LWL distinguishes $G_k$ and $H_k$, while the $\delta$-$(k-1)$-LWL does not distinguish $G_k$ and $H_k$.*

*Proof.* The fact that $\delta$-$k$-LWL distinguishes the graphs $G_k$ and $H_k$ follows from Lemma 28. We know argue that the $\delta$-$(k-1)$-LWL does not distinguish the two graphs. First, the (oblivious) $k$-WL has the same expressive power in distinguishing non-isomorphic graphs as the non-oblivious or folklore $(k-1)$-WL; see [Grohe (2021)] for details. Hence, it will not distinguish the graphs $G_k$ and $H_k$. The non-oblivious $(k-1)$-WL ([Grohe, 2021]) uses the following aggregation function

$$M^{(t)}((v_1, \ldots, v_{k-1})) \coloneqq \{\!\!\{(C_k^{(t)}(\theta_1(\mathbf{v}, w)), \ldots, C_k^{(t)}(\theta_{k-1}(\mathbf{v}, w))) \mid w \in V(G)\}\!\!\},$$

instead of Equation 8. Notice that from $(C_k^{(t)}(\theta_1(\mathbf{v}, w)), \ldots, C_k^{(t)}(\theta_{k-1}(\mathbf{v}, w)))$ we can recover if there is an edge between the vertex $w$ and a vertex $v_j$ for $j$ in $[k-1]$ in the underlying graph. Hence, the non-oblivious $(k-1)$-WL is at least as powerful as the $\delta$-$(k-1)$-LWL, implying that the $\delta$-$(k-1)$-LWL is weaker than the $\delta$-$k$-LWL. $\square$

We now construct non-isomorphic multi-relational graphs $G_r = (V(G_r), R_1(G_r), \ldots, R_r(G_r), \ell)$ and $H_r = (V(H_r), R_1(H_r), \ldots, R_r(H_r), \ell)$ that can be distinguished by the $(k+1)$-RLWL but not by the $k$-RLWL.

Let $V(G_r) \coloneqq V(G_k)$ and $V(H_r) \coloneqq V(H_k)$. Further, let $R_i(G_r) \coloneqq E(G_k)$ and $R_i(H_r) \coloneqq E(H_k)$ for $i$ in $[r]$. By a straightforward inductive argument it follows that $M_\delta^{(t)}(\mathbf{v}) = M_\delta^{(t)}(\mathbf{w})$ implies $M_r^{(t)}(\mathbf{v}) = M_r^{(t)}(\mathbf{w})$ for all $k$-tuples $\mathbf{v}$ and $\mathbf{w}$ in $V(G_k)^k$ or $V(H_k)^k$. This finishes the proof.

# D R-GCN

Additionally, we probe a modification of the R-GCN model with an MLP transformation (denoted as R-GCN+MLP) to facilitate parameter sharing between different relation-specific message propagations:

$$\boldsymbol{h}_{v,\text{R-GCN}}^{(t)} \coloneqq \sigma\Big(\boldsymbol{h}_{v,\text{R-GCN}}^{(t-1)} \cdot \boldsymbol{W}_0^{(t)} + \sum_{i \in [r]} \text{MLP}\Big(\sum_{w \in N_i(v)} \boldsymbol{h}_{w,\text{R-GCN}}^{(t-1)} \cdot \boldsymbol{W}_i^{(t)}\Big)\Big) \in \mathbb{R}^e.$$

This modification has a slightly higher count of learnable parameters.

# E CompGCN

The original CompGCN architecture proposed in [Vashishth et al. (2020)] considers directed graphs with self-loops, and uses an additional sum to differentiate between in-going, out-going, and self-loop edges, a degree-based normalization, and different weight matrices for these three cases, i.e.,

$$\boldsymbol{h}_{v,\text{CompGCN}}^{(t)} \coloneqq \sigma\Big(\boldsymbol{h}_{v,\text{CompGCN}}^{(t-1)} \boldsymbol{W}_0^{(t)} + \sum_{i \in [r]} \sum_{d \in D} \frac{1}{c_{v,w}} \sum_{w \in N_i^d(v)} \phi\big(\boldsymbol{h}_{w,\text{CompGCN}}^{(t-1)}, \boldsymbol{z}_i^{(t)}\big) \boldsymbol{W}_{1,d}^{(t)}\Big) \in \mathbb{R}^e,$$

where $D \coloneqq \{\text{in}, \text{out}\}$, representing in-going an out-going edges, respectively. Here, $N_i^d(v)$ is the restriction of $N_i^d(v)$ of $N_i(v)$ to in-going, out-going, and self-loop edges incident to the vertex $v$. Further, $c_{v,w} \coloneqq \sqrt{|N_i^d(v)| \cdot |N_i^d(w)|}$. The update of the previous vertex state is performed via the self-loop direction which we separate into the term $\boldsymbol{h}_{v,\text{CompGCN}}^{(t-1)} \boldsymbol{W}_0^{(t)}$ for the sake of a unified notation.

In the ablation studies, we probe the following modifications and combinations of those.

- CompGCN without normalization (*-norm*):

$$\boldsymbol{h}_{v,\text{CompGCN}}^{(t)} \coloneqq \sigma\Big(\boldsymbol{h}_{v,\text{CompGCN}}^{(t-1)} \boldsymbol{W}_0^{(t)} + \sum_{i \in [r]} \sum_{d \in D} \sum_{w \in N_i^d(v)} \phi\big(\boldsymbol{h}_{w,\text{CompGCN}}^{(t-1)}, \boldsymbol{z}_i^{(t)}\big) \boldsymbol{W}_{1,d}^{(t)}\Big) \in \mathbb{R}^e,$$

- CompGCN without direction-specific weights (*-dir*):

$$\boldsymbol{h}_{v,\text{CompGCN}}^{(t)} := \sigma\Big(\boldsymbol{h}_{v,\text{CompGCN}}^{(t-1)}\boldsymbol{W}_0^{(t)} + \sum_{i\in[r]}\frac{1}{c_{v,w}}\sum_{w\in N_i(v)}\phi\big(\boldsymbol{h}_{w,\text{CompGCN}}^{(t-1)}, \boldsymbol{z}_i^{(t)}\big)\boldsymbol{W}_1^{(t)}\Big) \in \mathbb{R}^e,$$

- CompGCN without relations update: $\mathbf{z}_i^{t+1} = \mathbf{z}_i^t$ (*-rp*).

As a composition function $\phi(h_w, \mathbf{z}_i)$ we probe several element-wise functions and an MLP:

- add: $\phi(\mathbf{h}_w, \mathbf{z}_i) = \mathbf{h}_w + \mathbf{z}_i$ – element-wise addition
- mult: $\phi(\mathbf{h}_w, \mathbf{z}_i) = \mathbf{h}_w * \mathbf{z}_i$ – element-wise multiplication (Hadamard product)
- rotate (Sun et al., 2019): $\phi(\mathbf{h}_w, \mathbf{z}_i) = \mathbf{h}_w \odot \mathbf{z}_i$ – rotation in complex space
- MLP: $\phi(\mathbf{h}_w, \mathbf{z}_i) = \text{MLP}([\mathbf{h}_w, \mathbf{z}_i])$ where $[\cdot]$ is column-wise concatenation

# F   Datasets and Hyperparameters

Statistics about the datasets are presented in Table 1. As neither of the datasets contain an explicit validation set, we retain a random 15% sample of train vertices for validation and use it to optimize hyperparameters.

**Table 1:** Vertex classification datasets statistics.

| Dataset | Vertices | Edges | Relations | Train vertices | Test vertices | Classes |
|---------|----------|-------|-----------|----------------|---------------|---------|
| AIFB | 8,285 | 29,043 | 45 | 140 | 36 | 4 |
| AM | 1,666,764 | 5,988,321 | 133 | 802 | 198 | 11 |

Final hyperparameters are listed in Table 2, the total parameter count for all trained models is presented in Table 3. Due to the size of the AM graph and identified stability of the initial vertex feature dimension, we only train models with dimension $d = 4$ on AM.

**Table 2:** Hyperparameters

| | AIFB | | | AM | | |
|---|---|---|---|---|---|---|
| | R-GCN | R-GCN + MLP | CompGCN | R-GCN | R-GCN + MLP | CompGCN |
| # Layers | 2 | 2 | 2 | 3 | 3 | 2 |
| LR | 0.001 | 0.001 | 0.001 | 0.03 | 0.03 | 0.03 |
| # epochs | 8,000 | 8,000 | 8,000 | 100 | 400 | 800 |
| Dropout | | 0.0 | | | 0.0 | |
| Optimizer | | | | Adam | | |
| Weight decay | | | | 0.0005 | | |

**Table 3:** Parameter count

| | AIFB | | | AM | | |
|---|---|---|---|---|---|---|
| dim | R-GCN | R-GCN + MLP | CompGCN | R-GCN | R-GCN + MLP | CompGCN |
| 2 | 1,092 | 1,144 | 262 | | | |
| 4 | 2,912 | 2,992 | 576 | 20,311 | 20,655 | 1,292 |
| 8 | 8,736 | 8,920 | 1,168 | | | |
| 16 | 29,120 | 29,704 | 3,636 | | | |
| 32 | 104,832 | 106,984 | 10,852 | | | |
| 64 | 396,032 | 404,392 | 36,036 | | | |
| 128 | 1,537,576 | 1,570,600 | 129,412 | | | |

