# OpenReview forum: "Weisfeiler and Leman Go Relational"
_logconference.io/LOG/2022/Conference — LoG 2022 Poster_

### Official Review · Reviewer_bjGn · 2022-10-20

**Overall Score:** 8
**Confidence:** 3

**Review:**


**Summary**. This paper investigated the expressive power of two multi-relational GNNs. The authors generalized the 1-WL into multi-relational graphs and identified their limitations in distinguishing nodes in multi-relational graphs. To improve these models, the authors proposed a new relational GNN architecture that is proven to be strictly more expressive. The theoretical results are empirically verified on node classification over small and large multi-relational graphs.

**Strong points**

1) The expressive power of existing multi-relational GNNs has not yet been explored. The paper is the first work in this direction.
2) The proposed multi-relational GNN is theoretical more expressive
3) Nice theoretical analysis.

**Weak points**

1) The theoretical results and empirical results are demonstrated on node classification. However, link prediction or relation prediction over knowledge graphs is quite different than node classification. Are the results still true on link/relation prediction?
2) Many theorems are proposed. However, how novel of the new theorems is unclear and must be clarified somewhere.
3) As mentioned in 1), the new GNN has not yet been evaluated on linking prediction over knowledge graphs.

**Questions**

1) There are many other improved RGCNs for inductive relation prediction such as GraIL [1], is your method still more expressive than GraIL? note that GraIL is able to learn the Horn logical rules.
2) For relation prediction, is your RGCN inductive like GraIL? note that the standard RGCN is not inductive in relation prediction but GraIL is.

[1] Komal K. Teru. et al. Inductive Relation Prediction by Subgraph Reasoning. ICML. 2020

Overall, I believe the idea is novel but it would highly increase the quality if the method could also be evaluated on linking prediction, or at least there should be some discussions. I would potentially increase my socre if the authors could address my concerns.

---

### Official Review · Reviewer_zxTN · 2022-10-21

**Overall Score:** 8
**Confidence:** 4

**Review:**

This paper presents theoretical results about the expressive power of two popular relational GNN models: R-GCN and CompGCN.
The main contribution is to analyze the equivalence between these models in terms of limited expressive power and suggesting extensions that are more expressive. In addition to the theoretical results, the authors also analyze numerically current implementations and relate them with the theory.

Strong points:

The paper considers an important problem, it is clearly written and technically correct. In particular, it provides interesting insights regarding the choice of composition functions. The presented work is overall well related with the existing literature. The numerical study is interesting and mostly convincing (see some points below).

Weak points:

The presented results do not appear very demanding and follow almost directly as a straightforward generalization of existing results between message-passing GNNs and k-WL. The relation between 1-RWL and 1-WL is not elaborated. This lowers down the significance/relevance of this work.

Globally, I believe the clarity in the presentation and its technical correctedness make this a solid contribution and the strong points outweigh the weak ones.

Comments:

C1- The authors introduce a variant of 1-WL (1-RWL) and mostly leverage existing literature to derive their results. Is 1-RWL really necessary to understand the expressive power of R-GCN and CompGCN? Is 1-RWL equivalent or 1-WL or not? Recent works on CSPs make the equivalence (up to syntactical differences) between 1-WL and relational structures without introducing variants of 1-WL, e.g.,

S. Butti and V. Dalmau. Fractional Homomorphism,  Weisfeiler-Leman invariance, and the Sherali-Adamshierarchy for the Constraint Satisfaction Problem. MFCS'21.

It would be interesting to understand this.

C2- The authors consider R-GCN and CompGCN. However, this choice of models is not motivated nor discussed. Why are no other architectures considered? It is difficult to assess the significance without this motivation.

C3- The numerical study appears a bit disconnected from the theory, e.g., dependencies on the dimension of the node features are not analyzed in the theoretical part. Also, the numerical performances are indeed very similar, but according to Fig.2 they are not strictly equivalent.

---

### Official Review · Reviewer_GBkz · 2022-10-21

**Overall Score:** 6
**Confidence:** 4

**Review:**

The authors study the expressive power of multi-relational GNNs, i.e., GNNs for graphs with multiple relation (edge) types, e.g., knowledge graphs. The authors consider two established architectures, namely the R-GCN and the Comp-GCN. In order to study the expressive power of these architectures, the authors derive the relational 1-Weisfeiler Lehman algorithm that considers the neighbours of a node per relation. By doing so, the authors show that both architectures have the same expressive power in terms of distinguishing non-isomorphic multi-relational graphs or distinguishing nodes. The authors also investigate under which composition functions is Comp-GCN most expressive (those that implement vector scaling). Finally, the authors propose an architecture that is more expressive, namely the local k-order relational networks (k-RNs), which they study by considering an extension of the local k Weisfeiler-Lehman algorithm, namely the multi-relational local k-WL.

The paper is well-written providing theoretical contributions to the community regarding the expressive power of multi-relational GNNs; an unexplored but relevant topic. Two main concerns I have with the paper is the k-RN architecture and the experiments performed. The authors present this new more expressive architecture that does not appear in the experiments section. Not sure why this is not the case, perhaps I am missing something. Instead, all the experiments are about Comp-GCN and R-GCN on 2 datasets with the authors aiming to show that their theoretical findings are reflected in practice. Moreover, there exist an ablation study for Comp-GCN and an evaluation of composition functions.

In my opinion, the experimental setting does not match the story of this paper. For example, I would be expecting the k-RN architecture to show up and being used. Moreover, I am not sure if the ablation studies and composition function evaluation are convincing or conclusive: is the community supposed to say "Hey, don't remove any part of Comp-GCN because that paper shows on two datasets that it performs worse". Given these observations, I would suggest to the authors to re-consider parts the experimental evaluation.

Further comments:
- I thought Lehman is spelled as Lehman and not Leman. However I see a lot of work where Leman is used. Not sure if this is correct.
- The introduction of the WL algorithm could be a bit structured, e.g., with a pseudocode
- Another interesting question that comes to my head is the behaviour of GNNs with an increasing number of relations
- While increasing k makes the algorithm more expressive, I would imagine that this is some sort of over-fitting (i.e., there must be a no free lunch here)
- What other examples are there for encoding labels as linearly independent vertex features besides one hot encoding?
- Page 6, there are r different relations and function atp accepts r as a subscript. I find this a bit confusing

---

### Official Review · Reviewer_TLWK · 2022-10-25

**Overall Score:** 6
**Confidence:** 3

**Review:**

## Summary

The paper deals with the study of the expressive power of Graph Neural Networks for multi-relational graph structured data. They take two GNN architectures i.e. Relational GCN (R-GCN) and Compositional GCN (CompGCN) and show their expressive power is equivalent to a modified version of 1-WL graph isomorphism test when initial features are linearly independent (R-GCN) and compositional functions used can express vector-scaling (CompGCN). Furthermore, the authors extend the higher-order GNNs in their local form i.e. local k-GNNs to the multi-relational data and propose corresponding k-RN neural architectures.

## Strengths:
1. The paper is well-written with good mathematical rigour. All the notations needed to understand the theory are well discussed in the Preliminaries section.
1. The proposition of k-RN strengthens the paper.
1. The paper gives some insights on the aspects specific to the architectures of R-GCN and CompGCN like the composition functions needed.

## Weaknesses:

1. The main drawback of this paper, seems to me, is the lack of new insights in the paper. The extension of 1-WL to multi-relational data does not seem very novel. In fact, 1-WL on multi-relational graphs is simply 1-WL on graphs with node features. We can just create additional node for each edge and it takes relation type as a node feature. We can then apply 1-WL on this graph. So, I don’t understand the need to study this as a separate problem.
1. Furthermore, it is not clear if formulating the problem of learning on multi-relational data in the form of graph isomorphism is well-motivated. For multi-relational data, more often we are not interested in inductive learning. Therefore, I believe it would be more apt to study the problem of expressivity from a different perspective which is more in line with the actual problems on such multi-relational data. However, this is not the main point, since there can be differences in approach.
1. The experimental section is lacking. Since, the paper is focused on expressivity, it would be ideal if there is a synthetic datasets with different levels of difficulty and the performance of R-GCN/CompGCN can be compared with 1-RWL as well as different architectural components. Also, it would good to see how would they compare with other 1-WL equivalent GNN with edge features like GINE. Additionally, k-RN is not tested experimentally.

In summary, the paper gives some insights on GNNs with multi-relational input data, specifically R-GCN and CompGCN architectures. However, generally the paper seems to be simple extension of other works on 1-WL/GNN equivalence and expressivity.

---

### Meta-Review · Area_Chair_pAqG · 2022-11-17

**Confidence:** 4
**Recommendation:** Accept

**Meta Review:**

The authors study the expressive power of relational GCNs (RGCN) and compositional GCN (compGCN) which are specific architectures for representing edge-attributed graphs. To study the expressiveness, the WL test is generalized to accomoadate edge types. The main theoretical result is that both methods are upper bounded by the relational version of the 1-WL test. A novel GNN layer for relational graphs is also proposed. All reviewers found the theoretical work is a strong contribution and merits acceptance on its own. Reviewers agreed that the experimental component requires future work but the current work is already valuable for the community.

---

### Decision · Program_Chairs · 2022-11-23

Accept (Poster)